# SPRINQL: Sub-optimal Demonstrations driven Offline Imitation Learning

**Huy Hoang**
Singapore Management University
`mh.hoang.2024@phdcs.smu.edu.sg`

**Tien Mai**
Singapore Management University
`atmai@smu.edu.sg`

**Pradeep Varakantham**
Singapore Management University
`pradeepv@smu.edu.sg`

## Abstract

We focus on offline imitation learning (IL), which aims to mimic an expert's behavior using demonstrations without any interaction with the environment. One of the main challenges in offline IL is the limited support of expert demonstrations, which typically cover only a small fraction of the state-action space. While it may not be feasible to obtain numerous expert demonstrations, it is often possible to gather a larger set of sub-optimal demonstrations. For example, in treatment optimization problems, there are varying levels of doctor treatments available for different chronic conditions. These range from treatment specialists and experienced general practitioners to less experienced general practitioners. Similarly, when robots are trained to imitate humans in routine tasks, they might learn from individuals with different levels of expertise and efficiency.

In this paper, we propose an offline IL approach that leverages the larger set of sub-optimal demonstrations while effectively mimicking expert trajectories. Existing offline IL methods based on behavior cloning or distribution matching often face issues such as overfitting to the limited set of expert demonstrations or inadvertently imitating sub-optimal trajectories from the larger dataset. Our approach, which is based on inverse soft-Q learning, learns from both expert and sub-optimal demonstrations. It assigns higher importance (through learned weights) to aligning with expert demonstrations and lower importance to aligning with sub-optimal ones. A key contribution of our approach, called SPRINQL, is transforming the offline IL problem into a convex optimization over the space of Q functions. Through comprehensive experimental evaluations, we demonstrate that the SPRINQL algorithm achieves state-of-the-art (SOTA) performance on offline IL benchmarks. Code is available at https://github.com/hmhuy0/SPRINQL.

## 1 Introduction

Reinforcement learning (RL) has established itself as a strong and reliable framework for sequential decision-making with applications in diverse domains: robotics [19, 18], healthcare [34, 28, 27], and environment generation [9, 24]. Unfortunately, RL requires an underlying simulator that can provide rewards for different experiences, which is usually not available.

Imitation Learning (IL) [16, 29, 13, 17] handles the lack of reward function by utilizing expert demonstrations to guide the learning scheme to compute a good policy. However, IL approaches still require the presence of a simulator that allows for online interactions. Initial works in Offline IL [35, 23, 40, 2] tackle the absence of simulator by considering an offline dataset of expert

demonstrations. These approaches extend upon Behavioral Cloning (BC), where we aim to maximize the likelihood of the expert's decisions from the provided dataset. The key advantage with BC is the theoretical justification on converging to expert behaviors given sufficient trajectories. However, when there are not enough expert trajectories, it often suffers from distributional shift issues [30]. Thus, a key drawback of these initial IL approaches is the need for a large number of expert demonstration datasets.

To deal with limited expert demonstrations, recent works utilize non-expert demonstration datasets to reduce the reliance on only expert demonstrations. These additional non-expert demonstrations are referred to as supplementary data. Directly applying BC to these larger supplementary datasets will lead to sub-optimal policies, so most prior work in utilizing supplementary data attempts to extract expert-like demonstrations from the supplementary dataset in order to expand the expert demonstrations [31, 21, 20, 37, 39]. These works assume expert-like demonstrations are present in the supplementary dataset and focus on identifying and utilizing those, while eliminating the non-expert demonstrations. Eliminating non-expert trajectories can result in loss of key information (e.g., transition dynamics) about the environment. Additionally, these works primarily rely on BC, which is known to overlook the sequential nature of decision-making problems – a small error can quickly accumulate when the learned policy deviates from the states experienced by the expert.

We develop our algorithm based on an inverse Q-learning framework that better captures the sequential nature of the decision-making [13] and can operate under the more realistic assumption that the data is collected from people/policies with lower expertise levels[1] (not experts). To illustrate, consider a scenario in robotic manipulation where the goal is to teach a robot to assemble parts. Expert demonstrations might show precise and efficient methods to assemble parts, but are limited in number due to the high cost and time associated with expert involvement. On the other hand, sub-optimal demonstrations from novice users are easier to obtain and more abundant. Our SPRINQL approach effectively integrates these sub-optimal demonstrations, giving appropriate weight to the expert demonstrations to ensure the robot learns the optimal assembly method without overfitting to the limited expert data or the inaccuracies in the sub-optimal data. We utilize these non-expert trajectories to learn a Q function that contributes to our understanding of the environment and the ground truth reward function.

**Contributions:** Overall, we make the following key contributions in this paper:
(i) We propose SPRINQL, a novel algorithm based on Q-learning for *__offline imitation learning with expert and multiple levels of sub-optimal demonstrations__*.
(ii) We provide key theoretical properties of the SPRINQL objective function, which enable the development of a scalable and efficient approach. In particular, we leverage distribution matching and reward regularization to develop an objective function for SPRINQL that not only help address the issue of limited expert samples but also utilizes non-expert data to enhance learning. Our objective function is not only convex within the space of $Q$ functions but also guarantees the return of a $Q$ function that lower-bounds its true value.
(iii) We provide an extensive empirical evaluation of our approach in comparison to existing best algorithms for offline IL with sup-optimal demonstrations. Our algorithms provide state-of-the-art (SOTA) performance on all the benchmark problems. Moreover, SPRINQL is able to recover a reward function that shows a high positive correlation with the ground-truth rewards, highlighting a unique advantage of our approach compared to other IL algorithms in this context.

## 1.1 Related Work

**Imitation Learning.** Imitation learning is recognized as a significant technique for learning from demonstrations. It begins with BC, which aims to maximize the likelihood of expert demonstrations. However, BC often under-performs in practice due to unforeseen scenarios [30]. To overcome this limitation, Generative Adversarial Imitation Learning (GAIL) [16] and Adversarial Inverse Reinforcement Learning (AIRL) [10] have been developed. These methods align the occupancy distributions of the policy and the expert within the Generative Adversarial Network (GAN) framework [14]. Alternatively, Soft Q Imitation Learning (SQIL) [29] bypasses the complexities of adversarial training by assigning a reward of +1 to expert demonstrations and 0 to the others, subsequently learning a value function based on these rewards. While the aforementioned imitation learning algorithms show

---

[1]While there have been works [5, 6, 7] that have attempted to minimize the required demonstrations using ranked datasets by preference-based RL, their algorithms are only applicable to online settings.

promise, they require interaction with the environment to obtain the policy distribution, which is often impractical.

**Offline Imitation Learning.**  ValueDICE [22] introduces a novel approach for off-policy training, suitable for offline training, using Stationary Distribution Corrections [25, 26]. However, ValueDICE necessitates adversarial training between the policy network and the $Q$ network, which can make the training slow and unstable. Recently, algorithms like PWIL [8] and IQ-learn [13] have optimized distribution distance, offering an alternative to adversarial training schemes. Since such approaches rely on occupancy distribution matching, a large expert dataset is often required to achieve the desired performance. Our approach, SPRINQL is able to bypass this requirement of a large set of expert demonstrations through the use of non-expert demonstrations (which are typically more available) in conjunction with a small set of expert demonstrations.

**Imitation Learning with Imperfect Demonstrations.**  T-REX [5] and D-REX [6] have shown that utilizing noise-ranked demonstrations as a reference-based approach can return a better policy without requiring expert demonstrations in online settings. Moreover, there are also several works [36, 33] that utilize the GAN framework [14] for sub-optimal datasets and have achieved several successes. Meanwhile, in the offline imitation learning context, TRAIL [38] utilizes sup-optimal demonstrations to learn the environment's dynamics. It employs a feature encoder to map the high-dimensional state-action space into a lower dimension, thereby allowing for a scalable way of learning of dynamics. This approach may face challenges in complex environments where predicting dynamics accurately is difficult, as shown in our experimental results. Other works assume that they can extract expert-like state-action pairs from the sub-optimal demonstration set and use them for BC with importance sampling [31, 37, 39, 21, 20]. However, expert-like state-actions might be difficult to accurately identify, as true reward information is not available. In contrast, our approach is more general, as we do not assume that the sub-optimal set contains expert-like demonstrations. We also allow for the inclusion of demonstrations of various qualities. Moreover, while prior works only recover policies, our approach enables the recovery of both expert policies and rewards, justifying the use of our method for Inverse Reinforcement Learning (IRL)[1].

## 2   Background

**Preliminaries.**  We consider a MDP defined by the following tuple $\mathcal{M} = \langle S, A, r, P, \gamma, s_0 \rangle$, where $S$ denotes the set of states, $s_0$ represents the initial state set, $A$ is the set of actions, $r : S \times A \to \mathbb{R}$ defines the reward function for each state-action pair, and $P : S \times A \to S$ is the transition function, i.e., $P(s'|s, a)$ is the probability of reaching state $s' \in S$ when action $a \in A$ is made at state $s \in S$, and $\gamma$ is the discount factor. In reinforcement learning (RL), the aim is to find a policy that maximizes the expected long-term accumulated reward $\max_\pi \{\mathbb{E}_{(s,a)\sim\rho_\pi}[r(s,a)]\}$, where $\rho_\pi$ is the occupancy measure of policy $\pi$: $\rho_\pi(s,a) = (1 - \gamma)\pi(a|s)\sum_{t=1}^{\infty} \gamma^t P(s_t = s|\pi)$.

**MaxEnt IRL**  The objective in MaxEnt IRL is to recover a reward function $r(s, a)$ from a set of expert demonstrations, $\mathcal{D}^E$. Let $\rho^E$ be the occupancy measure of the expert policy. The MaxEnt IRL framework [41] proposes to recover the expert reward function by solving

$$\max_r \min_\pi \ \{\mathbb{E}_{\rho^E}[r(s,a)] - (\mathbb{E}_{\rho_\pi}[r(s,a)] - \mathbb{E}_{\rho_\pi}[\log \pi(s,a)])\} \tag{1}$$

Intuitively, the aim is to find a reward function that achieves the highest difference between the expected value of the expert policy and the highest expected value among all other policies (computed through the min loop).

**IQ-Learn**  Given a reward function $r$ and a policy $\pi$, the soft Bellman equation is defined as $\mathcal{B}_r^\pi[Q](s,a) = r(s,a) + \gamma\mathbb{E}_{s'}[V^\pi(s')]$, where $V^\pi(s) = \mathbb{E}_{a\sim\pi(a|s)}[Q(s,a) - \log \pi(a|s)]$. The Bellman equation $\mathcal{B}_r^\pi[Q] = Q$ is contractive and always yields a unique Q solution [13]. In IQ-learn, they further define an inverse soft-Q Bellman operator $\mathcal{T}^\pi[Q] = Q(s,a) - \gamma\mathbb{E}_{s'}[V^\pi(s')]$. [13] show that for any reward function $r(a, s)$, there is a unique $Q^*$ function such that $\mathcal{B}_r^\pi[Q^*] = Q^*$, and for a $Q^*$ function in the $Q$-space, there is a unique reward function $r$ such that $r = \mathcal{T}^\pi[Q^*]$. This result suggests that one can safely transform the objective function of the *MaxEnt IRL* from $r$-space to the

Q-space as follows:

$$\max_{Q} \min_{\pi} \quad \Phi(\pi, Q) = \mathbb{E}_{\rho}[\mathcal{T}^{\pi}[Q](s,a))] - \mathbb{E}_{\rho_{\pi}}[\mathcal{T}^{\pi}[Q](s,a)] + \mathbb{E}_{\rho_{\pi}}[\log \pi(s,a)] \qquad (2)$$

which has several advantages; [13] show that $\Phi(\pi, Q)$ is convex in $\pi$ and linear in $Q$, implying that (2) always yields a unique saddle point solution. In particular, (2) can be converted into a maximization over the Q-space, making the training problem no longer adversarial.

## 3 SPRINQL

We now describe our inverse soft-Q learning approach, referred to as **SPRINQL** (**S**ub-o**P**timal demonstrations driven **R**eward regularized **IN**verse soft **Q** **L**earning). We first describe the three key components in the SPRINQL formulation:

(1) We formulate the objective function that enables matching the occupancy distribution of not just expert demonstrations, but also sub-optimal demonstrations.

(2) To mitigate the effect of limited expert samples (and larger sets of sub-optimal samples) that can bias the distribution matching of the first step to sub-optimal demonstrations, we introduce a *reward regularization term* within the objective. This regularization term is to ensure reward function allocates higher values to state-action pairs that appear in higher expertise demonstrations.

(3) We show that while this new objective does not have the same advantageous properties as the one in inverse Q-learning [13], with some minor (yet significant) changes it is possible to restore all the important properties.

### 3.1 Distribution Matching with Expert and Suboptimal Demonstrations

We consider a setting where there are demonstrations classified into several sets of different expertise levels $\mathcal{D}^1, \mathcal{D}^2, ...., \mathcal{D}^N$, where $\mathcal{D}^1$ consists of expert demonstrations and all the other sets contains sub-optimal ones. This setting is general than existing work in IL with sup-optimal demonstrations, which typically assumes that there are only two quality levels: expert and sub-optimal. Let $\mathcal{D} = \bigcup_{i \in [N]} \mathcal{D}^i$ be the union of all the demonstration sets and $\rho^1, ..., \rho^N$ be the occupancy measures of the respective expert policies. The ordering of expected values across different levels of expert policies would then be given by:

$$\mathbb{E}_{\rho^1}[r^*(s,a)] > \mathbb{E}_{\rho^2}[r^*(s,a)] > ... > \mathbb{E}_{\rho^N}[r^*(s,a)],$$

where $r^*(.,.)$ are the *ground-truth* rewards. Typically, the number of demonstrations in first level, $\mathcal{D}^1$ is significantly lower than those from other expert levels, i.e., $|\mathcal{D}^1| \ll |\mathcal{D}^i|$, for $i = 2, ..., N$ The MaxEnt IRL objective from Equation 1 can thus be adapted as follows:

$$\max_{r} \min_{\pi} \quad \sum_{i \in [N]} w_i \mathbb{E}_{\rho^i}[r(s,a)] - \mathbb{E}_{\rho_{\pi}}[r(s,a)] + \mathbb{E}_{\rho_{\pi}}[\log \pi(s,a)] \qquad (3)$$

where $w_i \geq 0$ is the weight associated with the expert level $i \in [N]$ and we have $w_1 > w_2 > ... > w_N$ and $\sum_{i \in [N]} w_i = 1$. There are two key intuitions in the above optimization: (a) Expert level $i$ accumulates higher expected values than expert levels greater than $i$; and (b) Difference in values accumulated by expert policies and the maximum of all other policies is maximized. The optimization term can be rewritten as:

$$\mathbb{E}_{\rho^U}[r(s,a)] - \mathbb{E}_{\rho_{\pi}}[r(s,a)] - \mathbb{E}_{\rho_{\pi}}[\log \pi(s,a)],$$

where $\rho^U = \sum_{i \in [N]} w_i \rho^i$. Here we note that the expected reward $\sum_{i \in [N]} w_i \mathbb{E}_{\rho^i}[r(s,a)]$ is empirically approximated by samples from the demonstration sets $\mathcal{D}^1, \mathcal{D}^2, ...., \mathcal{D}^N$. The number of demonstrations in the best demonstration set $\mathcal{D}^1$ (i.e. the set of expert demonstrations) is significantly *lower* when compared to other demonstration sets. So, an empirical approximation of $\mathbb{E}_{\rho^1}[r(s,a)]$ using samples from $\mathcal{D}^1$ would be inaccurate.

### 3.2 Regularization with Reference Reward

We create a reference reward based on the provided expert and sub-optimal demonstrations and utilize the reference function to compute a regularization term that is added to the objective of Equation 3.

Concretely, we define a *reference reward* function $\bar{r}(s, a)$ such that:

$$\bar{r}(s, a) > \bar{r}(s', a'), \forall (s, a) \in \mathcal{D}^1 \text{ and } (s', a') \notin \mathcal{D}^1 \text{ and}$$

$$\bar{r}(s, a) > \bar{r}(s', a'), \forall (s, a) \in \mathcal{D}^2 \text{ and } \forall (s', a') \notin \mathcal{D}^2 \cup \mathcal{D}^1 \text{ and so on}$$

The aim here is to assign higher rewards to demonstrations from higher expertise levels, and zero rewards to those that do not belong to provided demonstrations. We will discuss how to concretely estimate such reference reward values later.

We utilize this reference reward as part of the reward regularization term, which is added into the MaxEntIRL objective in (3) as follows:

$$\max_r \min_\pi \Big\{ \underbrace{\mathbb{E}_{\rho^U}[r(s, a)] - \mathbb{E}_{\rho_\pi}[r(s, a)] + \mathbb{E}_{\rho_\pi}[\log \pi(s, a)]}_{\text{Occupancy matching}} - \underbrace{\alpha \mathbb{E}_{\rho^U}[(r(s, a) - \bar{r}(s, a))^2]}_{\text{Reward regularizer}} \Big\} \quad (4)$$

where $\alpha > 0$ is a weight parameter for the reward regularizer term. With (4), the goal is to find a policy with an occupancy distribution that matches with the occupancy distribution of different expertise levels appropriately (characterized by the weights, $w_i$). Simultaneously, it ensures that the learning rewards are close to the pre-assigned rewards, aiming to guide the learning policy towards replicating expert demonstrations, while also learning from sub-optimal demonstrations.

### 3.3 Concave Lower-bound on Inverse Soft-Q with Reward Regularizer

Even though (4) can be directly solved to recover rewards, prior research suggests that transforming (4) into the Q-space will enhance efficiency. We delve into this transformation approach in this section. As discussed in Section 2, there is a one-to-one mapping between any reward function $r$ and a function $Q$ in the Q-space. Thus, the maximin problem in (4) can be equivalently transformed as:

$$\max_Q \min_\pi \Big\{ \mathcal{H}(Q, \pi) \stackrel{def}{=} \mathbb{E}_{\rho^U}[\mathcal{T}^\pi[Q](s, a))] - \mathbb{E}_{\rho_\pi}[\mathcal{T}^\pi[Q](s, a))] + \mathbb{E}_{\rho_\pi}[\log \pi(s, a)]$$
$$- \alpha \mathbb{E}_{\rho^U}[(\mathcal{T}^\pi[Q](s, a)) - \bar{r}(s, a))^2] \Big\} \quad (5)$$

where $r(s, a)$ is replaced by $\mathcal{T}^\pi[Q](s, a)$ and

$$\mathcal{T}^\pi[Q](s, a) = Q(s, a) - \gamma \mathbb{E}_{s'}[V^\pi(s')], \ V^\pi(s) = \mathbb{E}_{a \sim \pi(a|s)}[Q(s, a) - \log \pi(a|s)]$$

In the context of single-expert-level, [13] demonstrated that the objective function in the $Q$-space as given in Equation 2 is concave in $Q$ and convex in $\pi$, implying that the maximin problem always has a unique saddle point solution. Unfortunately, this property does not hold in our case.

**Proposition 3.1.** $\mathcal{H}(Q, \pi)$ *(as defined in Equation 5) is concave in Q but is not convex in $\pi$.*

In general, we can see that the first and second term of (6) are convex in $Q$, but the reward regularizer term, which can be written as

$$\alpha \mathbb{E}_{\rho^U} \Big[ (Q(s, a) - \bar{r}(s, a) - \mathbb{E}_{s' \sim P(\cdot|s, a)} \mathbb{E}_{a' \sim \pi(\cdot|s')} (Q(s', a') - \log \pi(s', a')))^2 \Big],$$

is not concave in $\pi$ (details are shown in the Appendix). The property indicated in Proposition 3.1 implies that the maximin problem within the $Q$-space $\max_Q \min_\pi J(Q, \pi)$ may not have a unique saddle point solution and would be more challenging to solve, compared to the original inverse IQ-learn problem.

Another key property of Equation 2 is with regards to the inner minimization problem over $\pi$, which yields a unique closed-form solution, enabling the transformation of the *max-min* problem into a non-adversarial concave maximization problem within the Q-space. The closed-form solution was given by $\pi^Q = \text{argmax}_\pi V^\pi(s)$ for all $s \in S$. Unfortunately, this result also does not hold with the new objective function in (6), as formally stated below:

**Proposition 3.2.** $\mathcal{H}(Q, \pi)$ *may not necessarily be minimized at $\pi^*$ such that $\pi^* = \text{argmax}_\pi V^\pi(s)$, for all $s \in S$.*

To overcome the above challenges, our approach involves constructing a more tractable objective function that is a lower bound on the objective of (6). Let us first define $\Gamma(Q) = \min_\pi \mathcal{H}(Q, \pi)$. We then look at the regularization term, which causes all the aforementioned challenges, and write:

$$(\mathcal{T}^\pi[Q](s,a)) - \bar{r}(s,a))^2 = (Q(s,a) - \bar{r}(s,a) - \mathbb{E}_{s'}[V^\pi(s')])^2$$
$$= (Q(s,a) - \bar{r}(s,a))^2 + (\mathbb{E}_{s'}[V^\pi(s')])^2 + 2(\bar{r}(s,a) - Q(s,a))\mathbb{E}_{s'}[V^\pi(s')]$$

We then take out the negative part of $(\bar{r}(s,a) - Q(s,a))$ using ReLU, and consider a slightly new objective function as follows:

$$\widehat{\mathcal{H}}(Q, \pi) \overset{def}{=} \sum_{i \in [N]} w_i \mathbb{E}_{\rho^i}[\mathcal{T}^\pi[Q](s,a))] - (\mathbb{E}_{\rho_\pi}[\mathcal{T}^\pi[Q](s,a))] - \mathbb{E}_{\rho_\pi}[\log \pi(s,a)])$$

$$- \alpha \mathbb{E}_{\rho^U}\left[(Q(s,a) - \bar{r}(s,a))^2 + (\mathbb{E}_{s'}V^\pi(s'))^2 + 2\text{ReLU}(\bar{r}(s,a) - Q(s,a))\mathbb{E}_{s'}V^\pi(s')\right] \quad (6)$$

Let $\widehat{\Gamma}(Q) = \min_\pi \widehat{\mathcal{H}}(Q, \pi))$. The proposition below shows that $\widehat{\Gamma}(Q)$ always lower-bounds $\Gamma(Q)$.

**Proposition 3.3.** *For any $Q \geq 0$, we have $\widehat{\Gamma}(Q) \leq \Gamma(Q)$ and $\max_Q \widehat{\Gamma}(Q) \leq \max_Q \Gamma(Q)$. Moreover, $\Gamma(Q) = \widehat{\Gamma}(Q)$ if $Q(s,a) \leq \bar{r}(s,a)$ for all $(s,a)$.*

We note that assuming $Q \geq 0$ is not restrictive, as if the expert's rewards $r^*(s,a)$ are non-negative (typically the case), then the *true* soft-Q function, defined as $Q^*(s,a) = \mathbb{E}[\sum_{s_t, a_t} \gamma^t(r^*(s,a) - \log \pi(s,a))|(s_0, a_0) = (s,a)]$, should also be non-negative, for any $\pi$. As $\widehat{\Gamma}(Q)$ provides a lower-bound approximation of $\Gamma(Q)$, maximizing $\widehat{\Gamma}(Q)$ over the $Q$-space would drive $\Gamma(Q)$ towards its maximum value. It is important to note that, given that the inner problem involves minimization, obtaining an upper-bound approximation function is easier. However, since the outer problem is a maximization one, an upper bound would not be helpful in guiding the resulting solution towards optimal ones. The following theorem indicates that $\widehat{\Gamma}(Q)$ is more tractable to use.

**Theorem 3.4.** *For any $Q \geq 0$, the following results hold: (i) The inner minimization problem $\min_\pi \widehat{\mathcal{H}}(Q, \pi)$ has a unique optimal solution $\pi^Q$ such that $\pi^Q = argmin_\pi V^\pi(s)$ for all $s \in S$ and $\pi^Q(a|s) = \frac{\exp(Q(s,a))}{\sum_a \exp(Q(s,a))}$, (ii) $\max_\pi V^\pi(s) = \log(\sum_a \exp(Q(s,a))) \overset{def}{=} V^Q(s)$, and (iii) $\widehat{\Gamma}(Q)$ is concave for $Q \geq 0$.*

The above theorem tells us that new objective $\widehat{\Gamma}(Q)$ has a closed form where $V^\pi(s)$ is replaced by $V^Q(s)$. Moreover $\widehat{\Gamma}(Q)$ is concave for all $Q \geq 0$. The concavity is particularly advantageous, as it guarantees that the optimization objective is well-behaved and has a unique solution $Q^*$ such that $(Q^*, \pi^{Q^*})$ form a unique saddle point of $\max_Q \min_\pi \widehat{\mathcal{H}}(Q, \pi)$. ***Thus, our tractable objective has all the nice properties that the original IQ-Learn objective had, while being able to work for the offline case with multiple levels of expert trajectories and our reward regularizer.***

## 3.4 SPRINQL Algorithm

Algorithm 1 provides the overall SPRINQL algorithm. We first estimate the reference rewards in lines 2-6 of Algorithm 1 and the overall process is described in Section 3.4.1. Before we proceed to the overall training, we have to estimate the weights, $w_i$ (associated with the ranked demonstration sets) employed in $\hat{\mathcal{H}}(Q, \pi)$. We provide a description of this estimation procedure in Section 3.4.2. Finally, to enhance stability and mitigate over-estimation issues commonly encountered in offline Q-learning, we employ a conservative version of $\hat{\mathcal{H}}(Q, \pi)$ in lines 8-15 of the algorithm and is described in Section 3.4.3. Some other practical considerations are discussed in the appendix.

---

**Algorithm 1 SPRINQL**: Inverse soft-Q Learning with Sub-optimal Demonstrations

---

**Require:** $(\mathcal{D}^1, \mathcal{D}^2, ..., \mathcal{D}^N), (w_1, w_2, ..., w_N), \bar{r}_\eta, Q_\psi, \pi_\theta$
1: # estimate reward reference function
2: **for** iteration $i...N_e$ **do**
3:      $d \leftarrow (d^1, d^1, ..., d^N) \sim (\mathcal{D}^1, \mathcal{D}^2, ..., \mathcal{D}^N)$
4:      from dataset $d$, calculate $\mathcal{L}(\bar{r}_\eta)$ by (7)
5:      $\eta \leftarrow \eta - \nabla_\eta \mathcal{L}(\bar{r}_\eta)$
6: **end for**
7: # train SPRINQL
8: **for** iteration $i...N$ **do**
9:      $d \leftarrow (d^1, d^1, ..., d^N) \sim (\mathcal{D}^1, \mathcal{D}^2, ..., \mathcal{D}^N)$
10:      # Update Q function
11:      from dataset $d$, calculate $\widehat{\mathcal{H}}^C(Q_\psi, \pi_\theta)$ by (8)
12:      $\psi \leftarrow \psi + \nabla_\psi \widehat{\mathcal{H}}^C(Q_\psi, \pi_\theta)$
13:      # Update policy for actor-critic
14:      $\theta \leftarrow \theta + \nabla \left[\mathbb{E}_{a \sim \pi_\theta(a|s)}^{s \sim d}[Q_\psi(s,a) - \ln(\pi_\theta(a|s))]\right]$
15: **end for**

---

### 3.4.1 Estimating the Reference Reward

We outline our approach to automatically infer the reference rewards $\overline{r}(s,a)$ from the ranked demonstrations. The general idea is to learn a function that assigns higher values to higher expert-level demonstrations. To achieve this, let us define $R(\tau) = \sum_{(s,a)\in\tau} \overline{r}(s,a)$ (i.e., accumulated reward of trajectory $\tau$). For two trajectories $\tau_i$, $\tau_j$, let $\tau_i \prec \tau_j$ denote that $\tau_i$ is lower in quality compared to $\tau_j$ (i.e., $\tau_i$ belongs to demonstrations from lower-expert policies, compared to $\tau_j$). We follow the Bradley-Terry model of preferences [4, 5] to model the probability $P(\tau_i \prec \tau_j)$ as $P(\tau_i \prec \tau_j) = \frac{\exp(R(\tau_j))}{\exp(R(\tau_i))+\exp(R(\tau_j))}$ and use the following loss function:

$$\min_{\overline{r}}\{\mathcal{L}(\overline{r}) = \sum_{i\in[N]} \sum_{(s,a),(s',a')\in\mathcal{D}^i} (\overline{r}(s,a) - \overline{r}(s',a'))^2 - \sum_{h,k\in[N],h>k,\tau_i\in\mathcal{D}^h,\tau_j\in\mathcal{D}^k} \ln P(\tau_i \prec \tau_j)\} \quad (7)$$

where the first term of $\mathcal{L}(\overline{r})$ serves to guarantee that the reward reference values for $(s,a)$ pairs within the same demonstration group are similar, and the second term aims to increase the likelihood that the accumulated rewards of trajectories adhere to the expert-level order. Importantly, it can be shown below that $\mathcal{L}(\overline{r})$ is convex in $\overline{r}$ (Proposition 3.5), making the learning well-behaved. In practice, one can model $\overline{r}(s,a)$ by a neural network of parameters $\theta$ and optimize $\mathcal{L}(\theta)$ over $\theta$-space.

**Proposition 3.5.** $\mathcal{L}(\overline{r})$ *is strictly convex in* $\overline{r}$.

### 3.4.2 Preference-based Weight Learning for $w_i$

Each weight parameter $w_i$ used in (4) should reflect the quality of the corresponding demonstration set $\mathcal{D}^i$, which can be evaluated by estimating the average expert rewards of these sets. Although this information is not directly available in our setting, the reward reference values discussed earlier provide a convenient and natural way to estimate them. This leads to the following formulation for inferring the weights $w_i$ from the ranked data: $w_i = \frac{\mathbb{E}_{(s,a)\sim D^i}[\overline{r}(s,a)]}{\sum_{j\in[N]}\mathbb{E}_{(s,a)\sim D^j}[\overline{r}(s,a)]}$.

### 3.4.3 Conservative soft-Q learning

Over-estimation is a common issue in offline Q-learning due to out-of-distribution actions and function approximation errors [11]. We also observe this in our IL context. To overcome this issue and enhance stability, we leverage the approach in [23] to enhance our inverse soft-Q learning. The aim is to learn a *conservative* soft-Q function that lower-bounds its true value. We formulate the *conservative inverse soft-Q* objective as:

$$\widehat{\mathcal{H}}^C(Q,\pi) = -\beta \sum_{s\sim\mathcal{D},\ a\sim\mu(a|s)} [Q(s,a)] + \widehat{\mathcal{H}}(Q,\pi) \quad (8)$$

where $\mu(a|s)$ is a particular state-action distribution. We note that in (8), the *conservative* term is added to the objective function, while in the conservative Q-learning algorithm [23], this term is added to the Bellman error objective of each Q-function update. This difference makes the theory developed in [23] not applicable. In Proposition 3.6 below, we show that solving $\max_Q \widehat{\mathcal{H}}^C(Q)$ will always yield a Q-function that is a lower bound to the Q function obtained by solving $\max_Q \widehat{\mathcal{H}}(Q,\pi)$.

**Proposition 3.6.** *Let* $\widehat{Q} = argmax_Q\widehat{\mathcal{H}}(Q,\pi)$ *and* $\widehat{Q}^C = argmax_Q\widehat{\mathcal{H}}^C(Q,\pi)$, *we have* $\sum_{s\sim\mathcal{D},a\sim\mu(a|s)} \widehat{Q}^C(s,a) \leq \sum_{s\sim\mathcal{D},a\sim\mu(a|s)} \widehat{Q}(s,a)$.

We can adjust the scale parameter $\beta$ in Equation 8 to regulate the conservativeness of the objective. Intuitively, if we optimize $Q$ over a lower-bounded Q-space, increasing the scale parameter $\beta$ will force each $Q(s,a)$ towards its lower bound. Consequently, when $\beta$ is sufficiently large, $\widehat{Q}^C$ will point-wise lower-bound $\widehat{Q}$, i.e., $\widehat{Q}^C(s,a) \leq \widehat{Q}(s,a)$ for all $(s,a)$.

## 4 Experiments

### 4.1 Experiment Setup

**Baselines.** We compare our SPRINQL with SOTA algorithms in offline IL with sub-optimal demonstrations: TRAIL [38], DemoDICE [21], and DWBC [37]. Moreover, we also compare with

other straightforward baselines: BC with only sub-optimal datasets (BC-O), BC with only expert data (BC-E), BC with all datasets (BC-both), and BC with fixed weight for each dataset (W-BC), SQIL [29], and IQ-learn [13] with only expert data. In particular, since TRAIL, DemoDICE, and DWBC were designed to work with only two datasets (one expert and one supplementary), in problems with multiple sub-optimal datasets, we combine all the sub-optimal datasets into one single supplementary dataset. Meanwhile, BC-E, BC-O, and BC-both combine all available data into a large dataset for learning, while W-BC optimizes $\sum_i \mathbb{E}_{s,a\sim\mathcal{D}^i}-w_i \ln(\pi(a|s))$, where $w_i$ are our weight parameters. T-REX [5] is a PPO-based IL algorithm that can work with ranked demonstrations. It is, however, not suitable for our offline setting, so we do not include it in the main comparisons. Nevertheless, we will later conduct an ablation study to compare SPRINQL with an adapted version of T-REX. Moreover, [15] developed IPL for learning from offline preference data. This approach requires comparisons between every pair of trajectories, thus is not suitable for our context.

**Environments and Data Generation:** We test on five Mujoco tasks [32] and four arm manipulation tasks from Panda-gym [12]. The maximum number transition of $(s, a)$ per trajectory is 1000 (or 1k for short) for the Mujoco and is 50 for Panda-gym tasks (descriptions in Appendix B.3). The sub-optimal demonstrations have been generated by randomly adding noise to expert actions and interacting with the environments. We generated large datasets of expert and non-expert demonstrations. For each seed, we randomly sample subsets of demonstrations for testing. This approach allows us to test with different datasets across seeds, rather than using fixed datasets for all seeds as in previous works. More details of these generated databases can be found in the Appendix B.4.

**Metric:** The return is normalized by $\texttt{score} = \frac{\texttt{return}-\texttt{R.return}}{\texttt{E.return}-\texttt{R.return}} \times 100$ where $\texttt{R.return}$ is the mean return of random policy and $\texttt{E.return}$ is the mean return of the expert policy. The scores are calculated by taking the last ten evaluation scores of each seed, with five seeds per report.

**Experimental Concerns.** Throughout the experiments, we aim to answer the following questions: (**Q1**) How does SPRINQL perform compared to other baselines? (**Q2**) How do the distribution matching and reward regularization terms impact the performance of SPRINQL? (**Q3**) What happens if we augment (or reduce) the expert data while maintaining the sub-optimal datasets? (**Q4**) What happens if we augment (or reduce) the sub-optimal data while maintaining the expert dataset? (**Q5**) How does the conservative term help in our approach? (**Q6**) How does increasing $N$ (the number of expertise levels) affect the performance of SPRINQL? (**Q7**) Does the preference-based weight learning approach provide good values for the weights $w_i$? (**Q8**) How does SPRINQL perform in recovering the ground-truth reward function?

## 4.2 Main Comparison Results

In this section, we provide comparison results to answer (**Q1**) with three datasets (i.e., $N = 3$). Additional comparison results for $N = 2$ can be found in the appendix. From lowest to highest

| | Mujoco | | | Panda-gym | | | |
|---|---|---|---|---|---|---|---|
| | Cheetah | Ant | Humanoid | Push | PnP | Slide | Avg |
| BC-E | -3.2±0.9 | 6.4±19.1 | 1.3±0.2 | 8.2±3.8 | 3.7±2.7 | 0.0±0.0 | 2.7 |
| BC-O | 14.2±2.9 | 35.2±20.1 | 10.6±6.3 | 8.8±4.5 | 3.9±2.7 | 0.1±0.3 | 12.1 |
| BC-both | 13.2±3.6 | 47.0±5.9 | 9.0±3.5 | 9.0±4.3 | 4.4±3.0 | 0.1±0.4 | 13.8 |
| W-BC | 12.9±2.8 | 47.3±6.4 | 19.6±19.0 | 8.8±4.3 | 3.7±2.8 | 0.0±0.0 | 15.4 |
| TRAIL | -4.1±0.3 | -4.7±1.9 | 2.6±0.6 | 11.7±4.0 | 7.8±3.7 | 1.7±1.8 | 3.9 |
| IQ-E | -3.4±0.6 | -3.4±1.3 | 2.4±0.6 | 26.3±10.9 | 18.1±12.5 | 0.1±0.4 | 6.7 |
| IQ-both | -6.1±1.4 | -58.2±0.0 | 0.8±0.0 | 8.3±3.9 | 3.8±3.3 | 0.0±0.2 | -8.6 |
| SQIL-E | -5.0±0.7 | -33.8±7.4 | 0.9±0.1 | 9.6±3.3 | 3.2±2.9 | 0.1±0.3 | -4.2 |
| SQIL-both | -5.6±0.5 | -58.0±0.4 | 0.8±0.0 | 8.2±3.8 | 3.3±2.3 | 0.1±0.3 | -12.6 |
| DemoDICE | 0.4±2.0 | 31.7±8.9 | 2.6±0.8 | 8.1±3.7 | 4.3±2.4 | 0.1±0.5 | 7.9 |
| DWBC | -0.2±2.5 | 10.4±5.0 | 3.7±0.3 | 36.9±7.4 | 25.0±6.3 | 11.6±4.4 | 14.6 |
| SPRINQL (ours) | **73.6±4.3** | **77.0±5.6** | **82.9±11.2** | **72.0±5.3** | **63.2±6.4** | **37.7±6.6** | **67.7** |

Table 1: Comparison results for three *Mujoco* and three *Panda-gym* tasks.

expertise levels, we randomly sample (25k-10k-1k) transitions for *Mujoco* tasks and (10k-5k-100) transitions for Panda-gym tasks for every seed (details of these three-level dataset are provided in Appendix B.4). Table 1 shows comparison results across 3 *Mujoco* tasks and 3 *Panda-gym* tasks (the full results for all the nine environments are provided in Appendix C.1). In general, SPRINQL significantly outperforms other baselines on all the tasks.

### 4.3 Ablation Study - No Distribution Matching and No Reward Regularizer

We aim to assess the importance of the distribution matching and reward regularizer terms in our objective (**Q2**). To this end, we conduct an ablation study comparing SPRINQL with two variants: (i) **noReg-SPRINQL**, derived by removing the reward regularizer term from (6), and (ii) **noDM-SPRINQL**, obtained by removing the distribution matching term from (6). Here, we note that the **noDM-SPRINQL** performs Q-learning using the reward reference function, this can viewed as an adaption of the T-REX algorithm [5] to our offline setting. The conservative Q-learning term is employed in the SPRINQL and the two variants to enhance stability. The comparisons for $N = 2$ and $N = 3$ on five Mujoco tasks are shown in Figure 1 (the full comparison results for all tasks are provided in the appendix). These results clearly show that SPRINQL outperforms the other variants, indicating the value of both terms in our objective function.

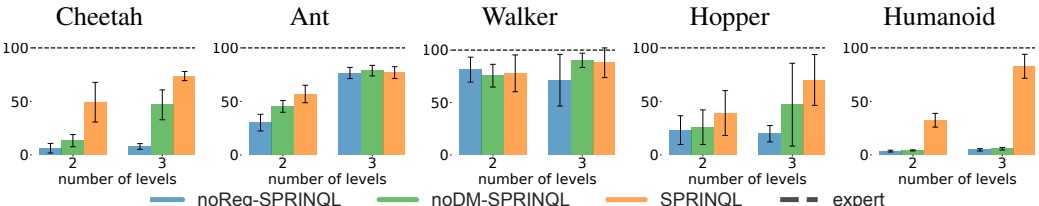

Figure 1: Comparison of three variants of SPRINQL across five Mujoco environments.

### 4.4 Other Experiments

Experiments addressing the other questions are provided in the appendix. Specifically, Sections C.1 and C.2 provide full comparison results for all the Mujoco and Panda-gym tasks for three and two datasets (i.e., $N = 3$ and $N = 2$), complementing the answer to **Q1**. Section C.3 provides the learning curves of the three variants considered in Section 4.3 above (answering **Q2**). Section C.4 provides experiments to answer **Q3** (*what would happen if we augment the expert dataset?*) and Section C.5 addresses **Q4** (*what would happen if we augment the sub-optimal dataset?*). Section C.6 experimentally shows the impact of the conservative term in our approaches (i.e., **Q5**). Section C.7 reports the performance of SPRINQL with varying numbers of expertise levels $N$ (i.e., **Q6**). Section C.8 addresses **Q7**, and Section C.10 shows how SPRINQL performs in terms of reward recovering (i.e. **Q8**). In addition, Section C.11 reports the distributions of the reference rewards and Section C.12 provides $\alpha$ choosing range.

Concretely, our extensive experiments reveal the following: (i) SPRINQL outperforms other baselines with two, three, or even larger numbers of datasets; (ii) the conservative term, distribution matching, and reward regularizer terms are essential to our objective—all three significantly contribute to the success of SPRINQL; (iii) the preference-based weight learning provides good estimates for the weights $w_i$; and (iv) SPRINQL performs well in recovering rewards, showing a high positive correlation with the ground-truth rewards, justifying the use of our method for IRL.

## 5 Conclusion and Limitations

**(Conclusion)** We have developed SPRINQL, a novel non-adversarial inverse soft-Q learning algorithm for offline imitation learning from expert and sub-optimal demonstrations. We have demonstrated that our algorithm possesses several favorable properties, contributing to its well-behaved, stable, and scalable nature. Additionally, we have devised a preference-based loss function to automate the estimation of reward reference values. We have provided extensive experiments based on several benchmark tasks, demonstrating the ability of our *SPRINQL* algorithm to leverage both expert and non-expert data to achieve superior performance compared to state-of-the-art algorithms.

**(Limitations)** Some limitations of this work include: (i) SPRINQL (and other baselines) still requires a large amount of sub-optimal datasets with well-identified expertise levels to learn effectively, (ii) there is a lack of theoretical investigation on how the sizes of the expert and non-expert datasets affect the performance of Q-learning, which we find challenging to address, and (iii) it lacks a theoretical exploration of how the reward regularizer term enhances the distribution matching term when expert samples are low—this question is relevant and interesting but also challenging to address. These limitations will pave the way for our future work.

## Acknowledgement

This research is supported by the National Research Foundation Singapore and DSO National Laboratories under the Al Singapore Programme (AISG Award No: AISG2-RP-2020-016) and Lee Kuan Yew Fellowship awarded to Pradeep Varakantham.

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

# A Missing Proofs

We provide proofs for the theoritical results claimed in the main paper.

**Proposition 3.1** $J(Q, \pi)$ *is concave in $Q$ but is not convex in $\pi$.*

*Proof.* We recall that

$$J(Q, \pi) = \sum_{i \in [N]} w_i \mathbb{E}_{\rho^i}[\mathcal{T}^\pi[Q](s, a))] - \mathbb{E}_{\rho_\pi}[\mathcal{T}^\pi[Q](s, a))] - \mathbb{E}_{\rho_\pi}[\log \pi(s, a)]$$

$$- \alpha \mathbb{E}_{\rho^U}[(\mathcal{T}^\pi[Q](s, a)) - \bar{r}(s, a))^2] \tag{9}$$

where $\mathcal{T}^\pi[Q](s, a)) = Q(s, a) - \gamma \mathbb{E}_{s' \sim P(s'|s, a)}[V^\pi(s')]$ and $V^\pi(s) = \mathbb{E}_{a \sim \pi(a|s)}[Q(s, a) - \log \pi(a|s)]$. We see that $\mathcal{T}^\pi[Q](s, a)$ is linear in $Q$ for any $(s, a)$. Thus, the first and second terms of $J(Q, \pi)$ in (9) are linear in $Q$. The last term of (9) involves a sum of squares of linear functions of $Q$, which are convex. So, $J(Q, \pi)$ is concave in $Q$.

To see that $J(Q, \pi)$ is generally not convex in $\pi$, we will consider a quadratic component of the reward regularization term $(\mathcal{T}^\pi[Q](s, a)) - \bar{r}(s, a))^2$ and show that there is an instance of $Q$ and $\bar{r}$ values that makes this term convex. We first write:

$$\mathcal{T}^\pi[Q](s, a)) - \bar{r}(s, a) = Q(s, a) - \gamma \sum_{s'} P(s'|s, a) V^\pi(s') - \bar{r}(s, a)$$

$$= Q(s, a) - \gamma \sum_{s'} P(s'|s, a) \sum_{a''} \pi(a''|s')(Q(s', a'') - \log \pi(a''|s')) - \bar{r}(s, a)$$

For simplification, let us choose $Q(s', a'') = 0$ for all $s'$ such that $P(s'|s, a) > 0$. This allows us to simplify $\mathcal{T}^\pi[Q](s, a)) - \bar{r}(s, a)$ as

$$\mathcal{T}^\pi[Q](s, a)) - \bar{r}(s, a) = \gamma \sum_{s'} P(s'|s, a) \sum_{a''} \pi(a''|s') \log \pi(a''|s') + Q(s, a) - \bar{r}(s, a)$$

We further see that, for any $s \in S$, $\sum_{a \in A} \pi(a|s) \log \pi(a|s)$ achieves its minimum value at $\pi(a|s) = 1/|A|$ for all $a \in |A|$, and $\sum_{a \in A} \pi(a|s) \log \pi(a|s) \geq \log 1/|A|$ for any policy $\pi$. As a result we have:

$$\gamma \sum_{s'} P(s'|s, a) \sum_{a''} \pi(a''|s') \log \pi(a''|s') \geq \gamma \log \frac{1}{|A|}$$

So if we select $Q(s, a)$ such that $Q(s, a) - \bar{r}(s, a) - \gamma \log |A| \geq 0$, then $\mathcal{T}^\pi[Q](s, a)) - \bar{r}(s, a) \geq 0$ for any $\pi$. Now we consider the quadratic function $\Gamma(\pi) = (\lambda(\pi))^2$ where $\lambda(\pi) = \mathcal{T}^\pi[Q](s, a)) - \bar{r}(s, a)$. Since each term $\pi(a'') \log \pi(a''|s')$ is convex in $\pi$, $\lambda(\pi)$ is convex in $\pi$. To show $\Gamma(\pi)$ is convex in $\pi$, we will show that for any two policies $\pi_1, \pi_2$ and $\alpha \in [0, 1]$, $\Gamma(\alpha \pi_1 + (1 - \alpha)\pi_2) \leq \alpha \Gamma(\pi_1) + (1 - \alpha)\Gamma(\pi)$. To this end, we write

$$\alpha \Gamma(\pi_1) + (1 - \alpha)\Gamma(\pi) \overset{(a)}{\geq} (\alpha \lambda(\pi_1) + (1 - \alpha)\lambda(\pi_2))^2$$

$$\overset{(b)}{\geq} (\lambda(\alpha \pi_1 + (1 - \alpha)\pi_2))^2$$

$$= \Gamma(\alpha \pi_1 + (1 - \alpha)\pi_2)$$

where $(a)$ is because the function $h(t) = t^2$ is convex in $t$, and $(b)$ is because

(i) $\alpha \lambda(\pi_1) + (1 - \alpha)\lambda(\pi_2) \geq \lambda(\alpha \pi_1 + (1 - \alpha)\pi_2)$ (as $\lambda(\pi)$ is convex in $\pi$)

(ii) $\alpha \lambda(\pi_1) + (1 - \alpha)\lambda(\pi_2)$ and $\lambda(\alpha \pi_1 + (1 - \alpha)\pi_2)$ are both non-negative, and function $h(t) = t^2$ is increasing for all $t \geq 0$.

So, we see that with the $Q$ values chosen above, function $(\mathcal{T}^\pi[Q](s, a)) - \bar{r}(s, a))^2)$ is convex and $-\alpha(\mathcal{T}^\pi[Q](s, a) - \bar{r}(s, a))^2$ is concave. So, intuitively, when $\alpha$ is sufficiently large, $J(Q, \pi)$ would be almost concave (so not convex), which is the desired result.

$\square$

**Proposition 3.2** $J(Q, \pi)$ *may not necessarily be minimized at* $\pi_Q$ *such that* $\pi_Q = argmax_\pi \, V^\pi(s)$, *for all* $s \in S$.

*Proof.* We first write $J(Q, \pi)$ as

$$J(Q, \pi) = \sum_{i \in [N]} w_i \mathbb{E}_{\rho^i}[\mathcal{T}^\pi[Q](s, a))] - \mathbb{E}_{\rho_\pi}[\mathcal{T}^\pi[Q](s, a))] - \mathbb{E}_{\rho_\pi}[\log \pi(s, a)]$$
$$- \alpha \mathbb{E}_{\rho^U}[(\mathcal{T}^\pi[Q](s, a)) - \bar{r}(s, a))^2]$$
$$= \sum_{i \in [N]} w_i \mathbb{E}_{\rho^i}[Q(s, a) - \gamma \mathbb{E}_{s'} V^\pi(s')] - (1 - \gamma) \mathbb{E}_{s_0} V^\pi(s_0)$$
$$- \alpha \mathbb{E}_{\rho^U}[(Q(s, a) - \bar{r}(s, a) - \gamma \mathbb{E}_{s'} V^\pi(s'))^2] \qquad (10)$$

We then see that the terms $\mathbb{E}_{\rho^i}[Q(s, a) - \gamma \mathbb{E}_{s'} V^\pi(s')]$ and $-\gamma \mathbb{E}_{s_0} V^\pi(s_0)$ are minimized (over $\pi$) when $V^\pi(s)$, for all $s$, are maximized. We will prove that it would not be the case for the last term. Let us choose $Q$ and $\bar{r}$ such that $Q(s, a) - \bar{r}(s, a) > \gamma \mathbb{E}_{s'} V^{\pi_Q}(s')$. We see that for any policy $\pi \neq \pi_Q$, we have

$$Q(s, a) - \bar{r}(s, a) > \gamma \mathbb{E}_{s'} V^{\pi_Q}(s') \geq \mathbb{E}_{s'} V^\pi(s')$$
$$\text{Thus, } Q(s, a) - \bar{r}(s, a) - V^\pi(s') \geq Q(s, a) - \bar{r}(s, a) - V^{\pi_Q}(s') > 0$$

which implies that

$$-\alpha \mathbb{E}_{\rho^U}[(Q(s, a) - \bar{r}(s, a) - \gamma \mathbb{E}_{s'} V^\pi(s'))^2] \leq -\alpha \mathbb{E}_{\rho^U}[(Q(s, a) - \bar{r}(s, a) - \gamma \mathbb{E}_{s'} V^{\pi_Q}(s'))^2]$$

So the last term of (10) would not be minimized at $\pi = \pi_Q$. In fact, in the above scenario, this last term will be maximized at $\pi = \pi_Q$. As a result, there is always $\alpha$ sufficiently large such that the last term significantly dominates the other terms and $J(Q, \pi)$ is not minimized at $\pi = \pi_Q$. $\qquad \square$

**Proposition 3.3** *For any* $Q \geq 0$, *we have* $\widehat{\Gamma}(Q) \leq \Gamma(Q)$ *and* $\max_{Q \geq 0} \widehat{\Gamma}(Q) \leq \max_{Q \geq 0} \Gamma(Q)$. *Mover,* $\Gamma(Q) = \widehat{\Gamma}(Q)$ *if* $Q(s, a) \leq \bar{r}(s, a)$ *for all* $s, a$.

*Proof.* We first write $\widehat{\mathcal{H}}(Q, \pi)$ as

$$\widehat{\mathcal{H}}(Q, \pi) = \sum_{i \in [N]} w_i \mathbb{E}_{\rho^i}[\mathcal{T}^\pi[Q](s, a))] - \mathbb{E}_{\rho_\pi}[\mathcal{T}^\pi[Q](s, a))] - \mathbb{E}_{\rho_\pi}[\log \pi(s, a)]$$

$$- \alpha \mathbb{E}_{\rho^U}\left[(Q(s, a) - \bar{r}(s, a))^2 + (\mathbb{E}_{s'} V^\pi(s'))^2 + 2\text{ReLU}(\bar{r}(s, a) - Q(s, a))\mathbb{E}_{s'} V^\pi(s)\right] \quad (11)$$

Since $Q \geq 0$, $V^\pi(s) = \mathbb{E}_{a \sim \pi(.|s)}[Q(s, a) - \log \pi(a|s)] \geq 0$. Thus $2\text{ReLU}(\bar{r}(s, a) - Q(s, a))\mathbb{E}_{s'} V^\pi(s) \geq 2(\bar{r}(s, a) - Q(s, a))\mathbb{E}_{s'} V^\pi(s)$. As a result, the last term of $\widehat{\mathcal{H}}(Q, \pi)$ is bounded as

$$- \alpha \mathbb{E}_{\rho^U}\left[(Q(s, a) - \bar{r}(s, a))^2 + (\mathbb{E}_{s'} V^\pi(s'))^2 + 2\text{ReLU}(\bar{r}(s, a) - Q(s, a))\mathbb{E}_{s'} V^\pi(s)\right]$$

$$\leq -\alpha \mathbb{E}_{\rho^U}\left[(Q(s, a) - \bar{r}(s, a))^2 + (\mathbb{E}_{s'} V^\pi(s'))^2 - 2(Q(s, a) - \bar{r}(s, a))\mathbb{E}_{s'} V^\pi(s)\right]$$

$$= -\alpha \mathbb{E}_{\rho^U}[(\mathcal{T}^\pi[Q](s, a) - \bar{r}(s, a))^2]$$

It then follows that $\widehat{\mathcal{H}}(Q, \pi) \leq J(Q, \pi)$. Thus, $\min_\pi \widehat{\mathcal{H}}(Q, \pi) \leq \min_\pi J(Q, \pi)$ or $\widehat{\Gamma}(Q) \leq \Gamma(Q)$. Mover, we see that if $\bar{r}(s, a) \geq Q(s, a)$ for all $(s, a)$, then $2\text{ReLU}(\bar{r}(s, a) - Q(s, a))\mathbb{E}_{s'} V^\pi(s) = 2(\bar{r}(s, a) - Q(s, a))\mathbb{E}_{s'} V^\pi(s)$, implying that $\widehat{\mathcal{H}}(Q, \pi) = J(Q, \pi)$ and $\widehat{\Gamma}(Q) = \Gamma(Q)$. This completes the proof.

$\qquad \square$

**Theorem 3.4** *For any $Q \geq 0$, the following results hold*

(i) *The inner minimization problem $\min_\pi \widehat{\mathcal{H}}(Q, \pi)$ has a unique optimal solution $\pi^*$ such that $\pi^Q = \text{argmin}_\pi V^\pi(s)$ for all $s \in S$ and*

$$\pi^Q(a|s) = \frac{\exp(Q(s,a))}{\sum_a \exp(Q(s,a))}.$$

(ii) $\max_\pi V^\pi(s) = \log(\sum_a \exp(Q(s,a))) \overset{def}{=} V^Q(s)$.

(iii) $\widehat{\Gamma}(Q)$ *is concave for $Q \geq 0$*

*Proof.* We first rewrite the formulation of $\widehat{\mathcal{H}}(Q, \pi)$ as

$$\widehat{\mathcal{H}}(Q, \pi) = \sum_{i \in [N]} w_i \mathbb{E}_{\rho^i}[Q(s,a) - \gamma \mathbb{E}_{s'} V^\pi(s')] - (1-\gamma)\mathbb{E}_{s_0} V^\pi(s_0)$$

$$- \alpha \mathbb{E}_{\rho^U}\left[(Q(s,a) - \bar{r}(s,a))^2 + (\mathbb{E}_{s'} V^\pi(s'))^2 + 2\text{ReLU}(\bar{r}(s,a) - Q(s,a))\mathbb{E}_{s'} V^\pi(s)\right]$$

We then see that the first and second term of $\widehat{\mathcal{H}}(Q, \pi)$ are minimized when $V^\pi(s)$ are minimized, i.e., at $\pi = \pi_Q$. For the last term, since $V^\pi(s) \geq 0$ (because $Q \geq 0$), $-(\mathbb{E}_{s'} V^\pi(s'))^2$ and $-2\text{ReLU}(\bar{r}(s,a) - Q(s,a))\mathbb{E}_{s'} V^\pi(s)$ are also minimized at $\pi = \pi_Q$. So, $\widehat{\mathcal{H}}(Q, \pi)$ is minimized at $\pi = \pi_Q$ as desired.

$(ii)$ is already proved in [13].

For $(iii)$, we rewrite $\widehat{\mathcal{H}}(Q, \pi)$ as

$$\widehat{\mathcal{H}}(Q, \pi) = \sum_{i \in [N]} w_i \mathbb{E}_{\rho^i}[\mathcal{T}^\pi[Q](s,a)] - (1-\gamma)\mathbb{E}_{s_0, a \sim \pi(a|s_0)}[Q(s_0, a) - \log \pi(a|s_0)]$$

$$- \alpha \mathbb{E}_{\rho^U}\left[(\bar{r}(s,a) - Q(s,a)) + \mathbb{E}_{s'} V^\pi(s))^2\right] + \alpha \mathbb{E}_{\rho^U}\left[(\min\{0, \bar{r}(s,a) - Q(s,a))\right]$$

$$= \sum_{i \in [N]} w_i \mathbb{E}_{\rho^i}[\mathcal{T}^\pi[Q](s,a)] - (1-\gamma)\mathbb{E}_{s_0, a \sim \pi(a|s_0)}[Q(s_0, a) - \log \pi(a|s_0)]$$

$$- \alpha \mathbb{E}_{\rho^U}\left[(\bar{r}(s,a) - Q(s,a)) + \mathbb{E}_{s'} V^\pi(s))^2\right] + \alpha \mathbb{E}_{\rho^U}\left[(\min\{0, \bar{r}(s,a) - Q(s,a))\right]$$

$$\tag{12}$$

Then, the first and second terms of (12) is linear in $Q$. The fourth term is concave. For third term, let $\Phi(Q) = |\bar{r}(s,a) - Q(s,a)| + \mathbb{E}_{s'} V^\pi(s)$. We see that $\Phi(Q) \geq 0$ for any $Q \geq 0$ and $\Phi(Q)$ is convex in $Q$ (because $V^\pi(s)$ is linear in $Q$). It then follows that, for any $\eta \in [0,1]$ and $Q_1, Q_2 \geq 0$, we have

$$\eta(\Phi(Q))^2 + (1-\eta)(\Phi(Q))^2 \overset{(a)}{\geq} (\eta\Phi(Q) + (1-\eta)\Phi(Q))^2$$

$$\overset{(b)}{\geq} (\Phi(\eta Q_1 + (1-\eta)Q_2))^2 \tag{13}$$

where $(a)$ is due to the fact function $h(t) = t^2$ is convex, and $(b)$ is because $h(t) = t^2$ is non-decreasing for all $t \geq 0$, and $\Phi(Q)$ is convex and always takes non-negative values. The last inequality in(13) implies that $(\Phi(Q))^2$ is convex in $Q$. So the last term of (12) is concave in $Q$. Putting all together we conclude that $\widehat{\mathcal{H}}(Q, \pi)$ is concave in $Q$ as desired. $\square$

**Proposition 3.5** $\mathcal{L}(\bar{r})$ *is strictly convex in $\bar{r}$.*

*Proof.* We first write $\mathcal{L}(\bar{r})$ as

$$\mathcal{L}(\bar{r}) = \sum_{i \in [N]} \sum_{(s,a),(s',a') \in \mathcal{D}^i} (\bar{r}(s,a) - \bar{r}(s',a'))^2 - \sum_{\substack{h,k \in [N], h<k \\ \tau_i \in \mathcal{D}^h, \tau_j \in \mathcal{D}^k}} \ln \frac{\exp(R(\tau_j))}{\exp(R(\tau_j)) + \exp(\tau_i)()} + \phi(\bar{r}) \tag{14}$$

$$= \sum_{i \in [N]} \sum_{(s,a),(s',a') \in \mathcal{D}^i} (\bar{r}(s,a) - \bar{r}(s',a'))^2 - \sum_{\substack{h,k \in [N], h<k \\ \tau_i \in \mathcal{D}^h, \tau_j \in \mathcal{D}^k}} \Big( R(\tau_j)$$

$$- \ln \left( \exp(R(\tau_j)) + \exp(R(\tau_i)) \right) + \phi(\bar{r}) \Big) \tag{15}$$

We then see that the first term is a sum of squares of linear functions of $\bar{r}$, thus is strictly convex. Moreover, since $R(\tau_i)$ is linear in $\bar{r}$ for any $\tau_i$ , the term $\ln(\exp(R(\tau_i)) + \exp(R(\tau_j)))$ has a log-sum-exp form. So this term is convex as well [3]. Putting all together we see that $\mathcal{L}(\bar{r})$ is strictly convex in $\bar{r}$ as desired. $\square$

**Proposition 3.6** *Let $\widehat{Q} = \mathrm{argmax}_Q \widehat{\mathcal{H}}(Q, \pi)$ and $\widehat{Q}^C = \mathrm{argmax}_Q \widehat{\mathcal{H}}^C(Q, \pi)$, we have*

$$\sum_{\substack{s \sim \mathcal{D} \\ a \sim \mu(a|s)}} \widehat{Q}^C(s,a) \leq \sum_{\substack{s \sim \mathcal{D} \\ a \sim \mu(a|s)}} \widehat{Q}(s,a)$$

*Proof.* We write

$$\widehat{\mathcal{H}}^C(\widehat{Q}^C, \pi) = -\beta \sum_{\substack{s \sim \mathcal{D} \\ a \sim \mu(a|s)}} \widehat{Q}^C(s,a) + \widehat{\mathcal{H}}(\widehat{Q}^C, \pi) \tag{16}$$

$$\overset{(a)}{\geq} -\beta \sum_{\substack{s \sim \mathcal{D} \\ a \sim \mu(a|s)}} \widehat{Q}(s,a) + \widehat{\mathcal{H}}(\widehat{Q}, \pi)$$

$$\overset{(b)}{\geq} -\beta \sum_{\substack{s \sim \mathcal{D} \\ a \sim \mu(a|s)}} \widehat{Q}(s,a) + \widehat{\mathcal{H}}(\widehat{Q}^C, \pi) \tag{17}$$

where $(a)$ is because $\widehat{Q}^C = \mathrm{argmax}_Q \widehat{\mathcal{H}}^C(Q, \pi)$ and $(b)$ is because $\widehat{\mathcal{H}}(\widehat{Q}, \pi) \geq \widehat{\mathcal{H}}(\widehat{Q}^C, \pi)$. Combine (16) and (17) we get

$$\beta \sum_{\substack{s \sim \mathcal{D} \\ a \sim \mu(a|s)}} \widehat{Q}^C(s,a) \leq \beta \sum_{\substack{s \sim \mathcal{D} \\ a \sim \mu(a|s)}} \widehat{Q}(s,a)$$

as desired. $\square$

## B  Additional Details

### B.1  Practical Training Objective

We discuss a practical implementation of the training objective in (4) using samples from both the expert and sub-optimal demonstration sets. We first note that the term $\mathbb{E}_{\rho_\pi}[\mathcal{T}^\pi[Q](s,a))] - \mathbb{E}_{\rho_\pi}[\log \pi(s,a)]$ can be written as [13]:

$$\mathbb{E}_{\rho_\pi}[\mathcal{T}^\pi[Q](s,a))] - \mathbb{E}_{\rho_\pi}[\log \pi(s,a)] = \mathbb{E}_{(s,s') \sim \rho^*}[V^\pi(s) - \gamma \mathbb{E}_{s'} V^\pi(s')]$$

for any valid occupancy measure $\rho^*$. So, for any policy occupancy measure, we can write this term though the union of expert policies $\rho^U$ as

$$\mathbb{E}_{\rho_\pi}[\mathcal{T}^\pi[Q](s,a))] - \mathbb{E}_{\rho_\pi}[\log \pi(s,a)] = \sum_{i \in [N]} w_i \mathbb{E}_{\rho^i}[V^\pi(s) - \gamma \mathbb{E}_{s'} V^\pi(s')] \tag{18}$$

As a result, by replacing $V^\pi(s)$ with $V^Q(s)$, we can write $\widehat{\Gamma}(Q)$ in a compact form as

$$\widehat{\Gamma}(Q) = \sum_{i \in [N]} w_i \mathbb{E}_{\rho^i}[Q(s,a) - \gamma \mathbb{E}_{s'} V^Q(s')]$$

$$- \sum_{i \in [N]} w_i \mathbb{E}_{\rho^i}[V^Q(s) - \gamma \mathbb{E}_{s'} V^Q(s')] - \alpha \sum_{i \in [N]} w_i \mathbb{E}_{\rho^i}[\Delta_{\overline{r}}^Q(s,a)] \qquad (19)$$

where $\Delta_{\overline{r}}^Q(s,a) = (Q(s,a) - \overline{r}(s,a))^2 + (\mathbb{E}_{s'} V^Q(s'))^2 + 2\text{ReLU}(\overline{r}(s,a) - Q(s,a))\mathbb{E}_{s'} V^Q(s')$.

In an empirical offline implementation, to maximize $\widehat{\Gamma}(Q)$, samples $(s,a,s')$ from demonstrations can be used to approximate the expectations over $\rho^i$: $\sum_{(s,a,s') \in \mathcal{D}^i}[Q(s,a) - \gamma V^Q(s')]$, $\sum_{(s,a,s') \in \mathcal{D}^i}[V^Q(s) - \gamma V^Q(s')]$, and $\sum_{(s,a) \in \mathcal{D}^i}[\Delta_{\overline{r}}^Q(s,a)]$.

We note that in continuous-action controls, the computation of $V^Q(s)$ involves a sum over infinitely many actions, which is impractical. In this case, we can update both $Q$ and $\pi$ in a soft actor-critic (SAC) manner. That is, for each $\pi$, we update $Q$ towards $\max_Q\{\widehat{\mathcal{H}}(Q,\pi)\}$ and for each $Q$, we update $\pi$ to bring it towards $\pi^Q$ by solving $\max_\pi\{V^\pi(s)\}$, for all $s$. As shown above, $\widehat{\mathcal{H}}(Q,\pi)$ in concave in $Q$ and $V^\pi(s)$ is convex in $\pi$, so we can expect this SAC will exhibit good behavior and stability.

## B.2 Algorithm Overview

The Figure 2 provide the overview of our algorithm.

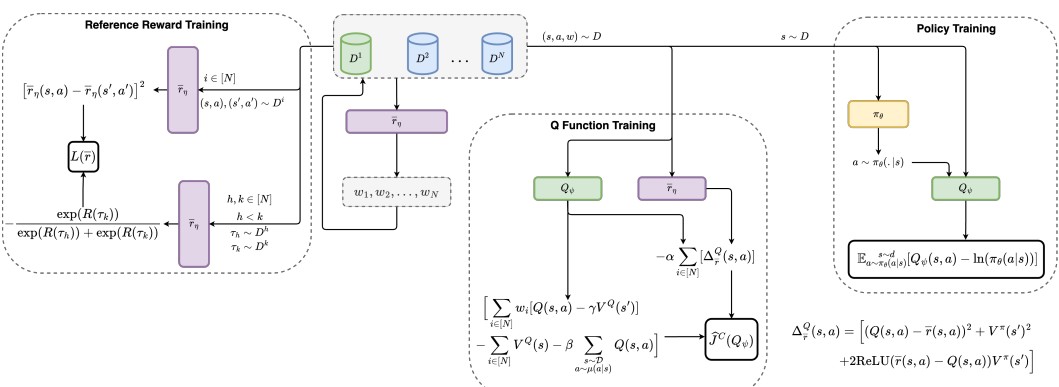

Figure 2: Overview of SPRINQL.

## B.3 Environments

This section provide a detailed description of the enviroments used in our experiments.

### B.3.1 Mujoco

MuJoCo gym environments [32] like HalfCheetah, Ant, Walker2d, Hopper, and Humanoid are integral to the field of reinforcement learning (RL), particularly in the domain of continuous control and robotics:

- `HalfCheetah`: This environment simulates a two-dimensional cheetah-like robot. The objective is to make the cheetah run as fast as possible, which involves learning complex, coordinated movements across its body.

- `Ant`: This environment features a four-legged robot resembling an ant. The challenge is to control the robot to move effectively, balancing stability and speed.

- `Walker2d`: This environment simulates a two-dimensional bipedal robot. The goal is to make the robot walk forward as fast as possible without falling over.

- `Hopper`: The Hopper environment involves a single-legged robot. The primary challenge is to balance and hop forward continuously, which requires maintaining stability while in motion.
- `Humanoid`: The Humanoid environment is among the most complex, featuring a bipedal robot with a human-like structure. The task involves mastering various movements, from walking to more complex maneuvers, while maintaining balance.

An expert is trying to maximize the reward function by moving with a trajectory length limit of 1000. All five environments are shown in Figure 3.

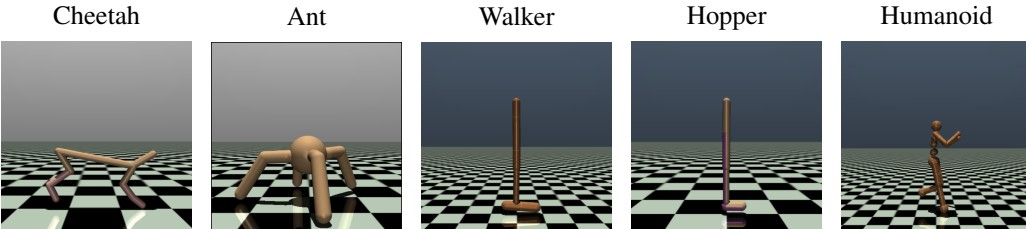

Figure 3: Five different Mujoco environments.

### B.3.2 Panda-gym

The *Panda-gym* environments [12], designed for the Franka Emika Panda robot in reinforcement learning, include *PandaReach*, *PandaPush*, *PandaPickandPlace*, and *PandaSlide*. Here's a short description of each:

- `PandaReach`: The task is to move the robot's gripper to a randomly generated target position within a specific volume. It focuses on precise control of the gripper's movement.
- `PandaPush`: In this environment, a cube placed on a table must be pushed to a target position. The gripper remains closed, emphasizing the robot's ability to manipulate objects through pushing actions.
- `PandaPickandPlace`: This more complex task involves picking up a cube and placing it at a target position above the table. It requires coordinated control of the robot's gripper for both lifting and accurate placement.
- `PandaSlide`: Here, the robot must slide a flat object (like a hockey puck) to a target position on a table. The gripper is fixed in a closed position, and the task demands imparting the right amount of force to slide the object to the target.

In these environments, the reward function is -1 for every time step it has not finished the task. Moreover, the maximum horizon is extremely short, with a maximum of 50, while an expert can complete the task after several steps. All four different environments are shown in Figure 4.

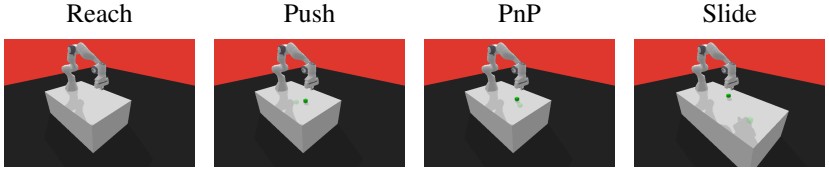

Figure 4: Five different Panda-gym environments.

### B.4   Qualities of the Generated Expert and Non-Expert Datasets

In this paper, we provide a new setting utilizing ranked sub-optimal datasets to perform imitation learning in the offline setting (illustration in Figure 5).

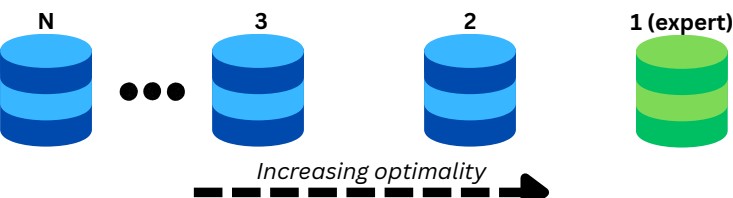

**Ranked sub-optimal datasets**

*Increasing optimality*

Figure 5: Training datasets illustration.

We create sub-optimal datasets by adding noise to actions of the expert policy and let them interact with the environments to collect the sub-optimal trajectories. Each task has its own difficulty and sensitivity to the noise of the actions. The averaged returns of the generated datasets, computed as percentages w.r.t. the maximum returns, are reported in Figure 6.

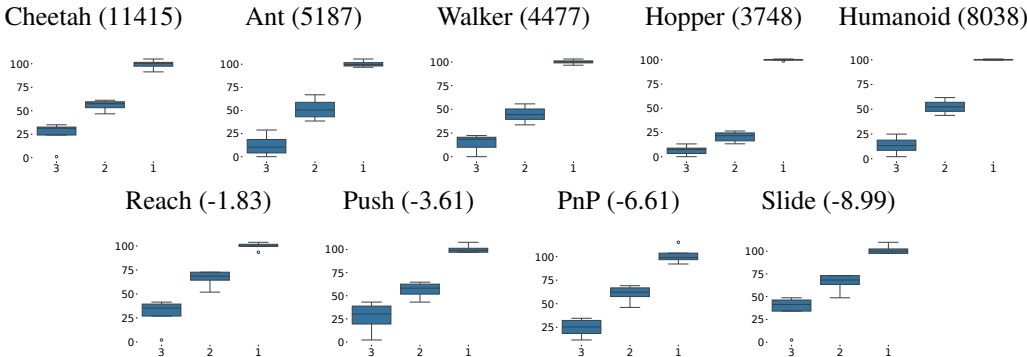

Figure 6: Whisker plots illustrate the average returns of both expert and non-expert datasets nine distinct environments in Mujoco and Panda-gym. The numerical values following the task names represent the actual mean return of the expert policy.

## B.5 Hyper Parameters and Experimental Implementations

- In our experiments, for every algorithm, we run five different seeds corresponding to five different datasets sampled from our databases in Appendix B.4.

- We use double Q critic network for our implementation to increase the stability in the offline training scheme.

- For IQ-learn [13],SQIL [29], we conduct experiment from its **official implement** with double Q critic network.

- For DemoDICE [21] we conduct experiment from its **official implementation**.

- For DWBC [37] we conduct experiment from its **official implementation**.

- Inspired by double Q-learning, to avoid overfitting in the offline setting, some experiments apply a training trick using KL-divergence between the target actor and the training actor to prevent rapid policy changes.

- For SAC-based algorithms, we use a fixed exploration parameter which is commonly used in previous work.

- We conducted all experiments on a total of 8 NVIDIA RTX A5000 GPUs and 64 core CPUs. We use 1 GPUs and 8 core CPUs per task with approximately one day per 5 seeds. The detailed hyper-parameters are reported in Table 2.

| HYPER PARAMETER | BC-BASED | SPRINQL |
|---|---|---|
| ACTOR NETWORK | [256,256] | [256,256] |
| CRITIC NETWORK | [256,256] | [256,256] |
| TRAINING STEP | 1,000,000 | 1,000,000 |
| GAMMA | 0.99 | 0.99 |
| LR ACTOR | 0.0001 | 0.00003 |
| LR CRITIC | 0.0003 | 0.0003 |
| LR REWARD REFERENCE | 0.0003 | 0.0003 |
| BATCH SIZE | 256 | 256 |
| SOFT UPDATE CRITIC FACTOR | - | 0.005 |
| SOFT UPDATE ACTOR FACTOR | - | 0.00003 |
| EXPLORATION TEMPERATURE | - | 0.01 |
| REWARD REGULARIZE TEMPERATURE ($\alpha$) | - | 1.0 |
| CQL TEMPERATURE ($\beta$) | - | 1.0 |

Table 2: Hyper parameters.

## C Supplementary Experiments

In this section, we present additional experiments that complement those reported in the main paper, along with ablation studies to address the experimental questions stated therein.

### C.1 Full Experiment Results for Mujoco and Panda-gym

We report the full results for the 5 different Mujoco environments and 4 different Panda-gym environments, with 3 datasets (i.e., $N = 3$), supplementing the results reported in Table 1 in the main paper. The detailed results for Mujoco in Table 3 and Panda-gym in Table 4

| | Cheetah | Ant | Walker | Hopper | Humanoid | Avg |
|---|---|---|---|---|---|---|
| BC-E | -3.2±0.9 | 6.4±19.1 | 0.3±0.4 | 8.3±4.1 | 1.3±0.2 | 2.6 |
| BC-O | 14.2±2.9 | 35.2±20.1 | 66.1±10.8 | 16.7±4.0 | 10.6±6.3 | 29.0 |
| BC-both | 13.2±3.6 | 47.0±5.9 | 63.9±7.3 | 22.6±12.8 | 9.0±3.5 | 31.1 |
| W-BC | 12.9±2.8 | 47.3±6.4 | 58.6±9.1 | 20.9±6.6 | 19.6±19.0 | 31.9 |
| TRAIL | -4.1±0.3 | -4.7±1.9 | 0.2±0.3 | 1.1±0.5 | 2.6±0.6 | -1.0 |
| IQ-E | -3.4±0.6 | -3.4±1.3 | 0.1±0.7 | 0.3±0.2 | 2.4±0.6 | -0.8 |
| IQ-both | -6.1±1.4 | -58.2±0.0 | -0.2±0.1 | 0.0±0.0 | 0.8±0.0 | -12.7 |
| SQIL-E | -5.0±0.7 | -33.8±7.4 | 0.2±0.2 | 0.2±0.0 | 0.9±0.1 | -7.5 |
| SQIL-both | -5.6±0.5 | -58.0±0.4 | -0.2±0.0 | 0.0±0.0 | 0.8±0.0 | -12.6 |
| DemoDICE | 0.4±2.0 | 31.7±8.9 | 7.2±3.1 | 18.7±7.5 | 2.6±0.8 | 12.1 |
| DWBC | -0.2±2.5 | 10.4±5.0 | 25.1±16.3 | 66.0±20.9 | 3.7±0.3 | 21.0 |
| SPRINQL (ours) | **73.6±4.3** | **77.0±5.6** | **87.9±14.2** | **70.0±23.8** | **82.9±11.2** | **78.3** |

Table 3: Comparison results for *Mujoco* tasks.

|         | Reach      | Push       | PnP        | Slide     | Avg  |
|---------|-----------|-----------|-----------|-----------|------|
| BC-E    | 13.9±5.9  | 8.2±3.8   | 3.7±2.7   | 0.0±0.0   | 6.5  |
| BC-O    | 16.2±5.5  | 8.8±4.5   | 3.9±2.7   | 0.1±0.3   | 7.3  |
| BC-both | 16.3±5.0  | 9.0±4.3   | 4.4±3.0   | 0.1±0.4   | 7.3  |
| W-BC    | 15.7±5.1  | 8.8±4.3   | 3.7±2.8   | 0.0±0.0   | 7.0  |
| TRAIL   | 18.3±5.1  | 11.7±4.0  | 7.8±3.7   | 1.7±1.8   | 9.9  |
| IQ-E    | 97.7±2.4  | 26.3±10.9 | 18.1±12.5 | 0.1±0.4   | 35.6 |
| IQ-both | 5.7±3.4   | 8.3±3.9   | 3.8±3.3   | 0.0±0.2   | 4.5  |
| SQIL-E  | 22.1±15.1 | 9.6±3.3   | 3.2±2.9   | 0.1±0.3   | 8.8  |
| SQIL-both | 8.0±4.2 | 8.2±3.8   | 3.3±2.3   | 0.1±0.3   | 4.9  |
| DemoDICE | 14.0±5.3 | 8.1±3.7   | 4.3±2.4   | 0.1±0.5   | 6.6  |
| DWBC    | 93.4±4.3  | 36.9±7.4  | 25.0±6.3  | 11.6±4.4  | 41.7 |
| SPRINQL (ours) | **99.8±0.9** | **72.0±5.3** | **63.2±6.4** | **37.7±6.6** | **68.2** |

Table 4: Comparison results for *Panda-gym* tasks.

## C.2 Comparison Results for $N = 2$, i.e., One Expert and One Sub-optimal Datasets - Q1

We provide additional comparison results for $N = 2$, complementing to the answer of (**Q1**), i.e., *how does SPRINQL perform compared to other baselines?* In this experiment, we want to test the ability of our algorithm in the same scenario of DemoDICE, DWBC which include expert dataset and only one supplementary dataset. Number of expert transitions in Mujoco domains is 1000 and 100 for Panda-gym. Meanwhile, for the supplementary dataset, it is 25000 transitions for Mujoco and 5000 for Panda-gym. In general, although we experience a downgrade in performance due to lack of ranking (only one supplementary dataset), leading to misunderstanding the true reward function, our method is still able to leverage the sub-optimal dataset for understanding the expert demonstrations and provide the highest average score. Reported results are shown in Table 5 6 and Figure 7.

| Method | Cheetah | Ant | Walker | Hopper | Humanoid | Avg |
|--------|---------|-----|--------|--------|----------|-----|
| BC | 3.4±1.3 | 39.2±6.6 | 45.5±14.4 | 28.4±9.0 | 24.4±25.2 | 28.2 |
| W-BC | 3.1±1.0 | 39.6±7.3 | 46.1±12.3 | 29.5±9.0 | 28.6±27.8 | 29.4 |
| DemoDICE | -1.6±0.6 | 31.6±7.5 | 6.5±1.7 | 31.9±12.3 | 2.7±0.6 | 14.2 |
| DWBC | -1.2±1.3 | 9.5±3.4 | 13.1±4.7 | **87.0±22.2** | 4.2±0.4 | 22.5 |
| SPRINQL (ours) | **49.2±18.5** | **56.9±8.2** | **77.8±17.6** | 39.1±21.0 | **32.4±6.4** | **51.1** |

Table 5: Comparison results for *Mujoco* tasks in two expert datasets.

| Method | Reach | Push | PnP | Slide | Avg |
|--------|-------|------|-----|-------|-----|
| BC | 17.1±4.8 | 8.1±3.8 | 3.6±2.4 | 0.3±0.6 | 7.3 |
| W-BC | 18.5±5.2 | 9.8±4.5 | 3.4±3.1 | 0.1±0.4 | 8.0 |
| DemoDICE | 14.8±5.7 | 8.8±4.4 | 3.8±2.9 | 0.2±0.6 | 6.9 |
| DWBC | **98.2±2.4** | 38.4±7.4 | 27.7±8.1 | 14.5±4.3 | 44.7 |
| SPRINQL (ours) | 93.9±3.3 | **61.3±7.2** | **64.2±6.7** | **26.6±5.5** | **61.5** |

Table 6: Comparison results for *Panda-gym* tasks in two expert datasets.

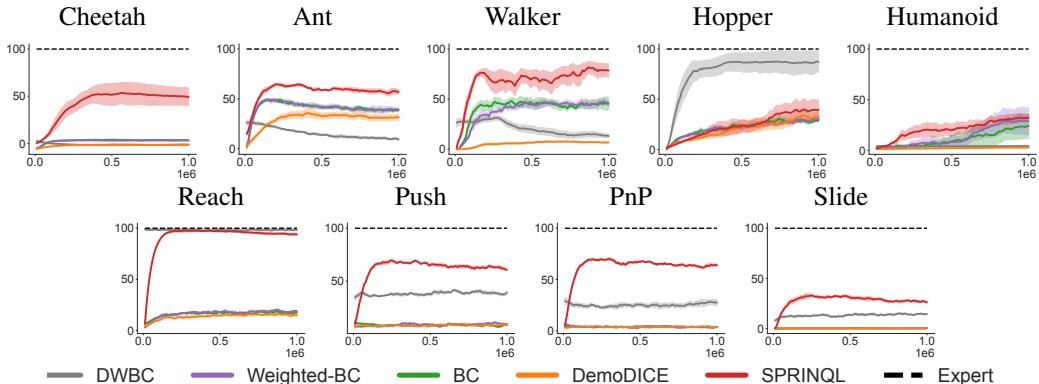

Figure 7: Learning curves of two dataset experiment.

## C.3 Learning Curves of noReg-SPRINQL and noDM-SPRINQL

This section reports additional results for the comparison between **SPRINQL** and the two variants **noReg-SPRINQL** and **noDM-SPRINQL** (supplementing the answer of **Q2** in the main paper). The comparison results for Panda-gym environments are reported in Figure 8 and the learning curves are plotted in Figure 9 and Figure 10.

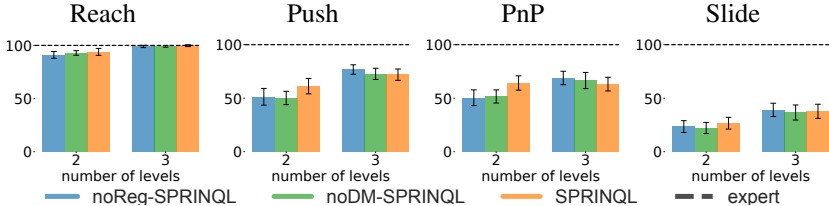

Figure 8: Ablation study show the performance of three variants of SPRINQL across four Panda-gym environments.

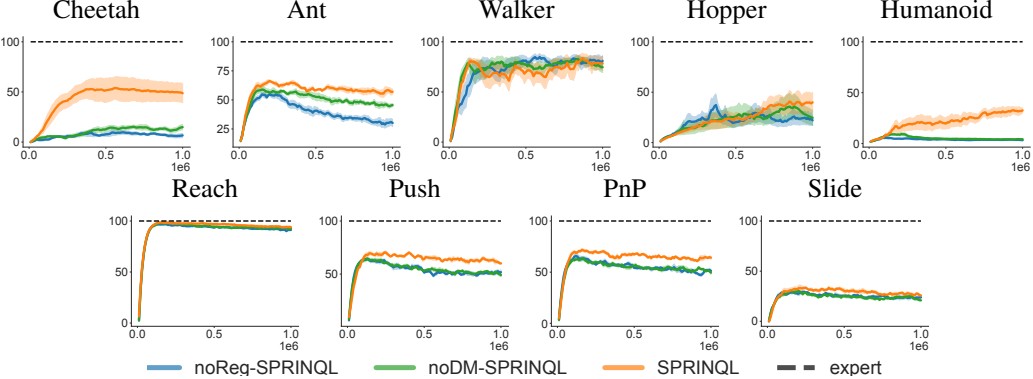

Figure 9: Two term ablation study for two level dataset scenario.

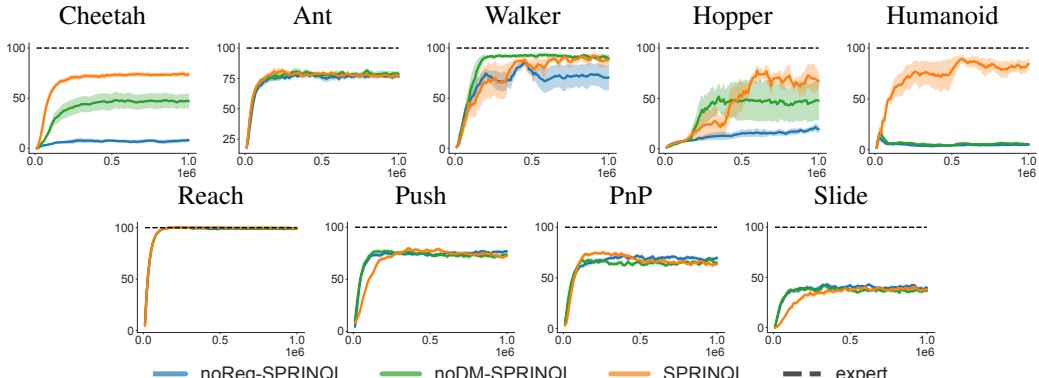

Figure 10: Two term ablation study for three level dataset scenario.

## C.4 Augmented Expert Demonstrations

We provide experiments to address (**Q3**) – *What happens if we augment the expert data while maintaining the sub-optimal datasets*? To this end, we use two *Mujoco* and two *Panda-gym* tasks, keeping the same sup-optimal datasets and add more expert demonstrations to the training sets. The comparison results are reported in Figure 11. For the Mujoco tasks, which are more difficult, adding more expert trajectories significantly enhances the performance of all the algorithms. However, for the two *Panda-gym* tasks, the influence of adding more expert data appears to be less significant in Push and completely absent in PnP. This would be because expert trajectories in these tasks are typically short, consisting of only 2-7 transitions. Hence, a larger quantity of additional expert data may be required to enhance performance.

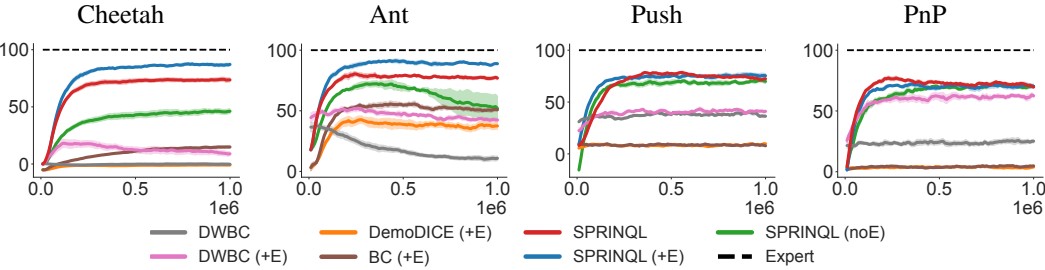

Figure 11: Comparison results with additional expert demonstrations; (+E) signifies that the expert dataset is increased from 1k to 5k for *Mujoco* and from 100 to 500 for *Panda-gym*, while (noE) indicates no expert data.

## C.5 Augmented Sup-optimal Demonstrations

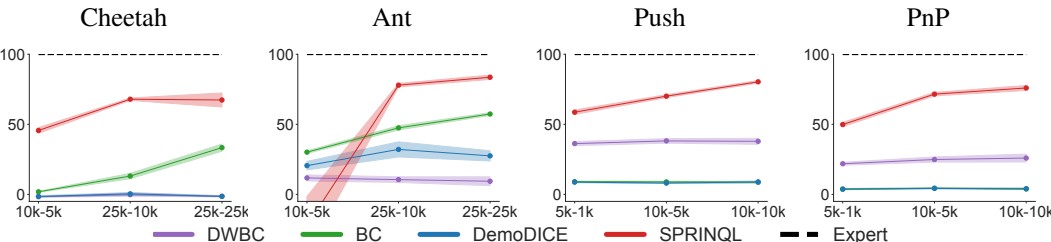

Figure 12: Performance comparisons with varied sub-optimal data sizes. The $x$-axis shows the size of Level 2 and 3 sup-optimal datasets, and $y$-axis shows the scores.

In this experiment, we want to answer the question (**Q4**) – *What happens if we augment (or reduce) the sub-optimal data while maintaining the expert dataset?*. We present numerical results to assess

the impact of sub-optimal data on the performance of our SPRINQL and other baselines. To this end, we keep the same expert dataset and adjust the non-expert data used in Table 1. The performance is reported in Fig.12. It is evident that reducing the amount of data in sub-optimal datasets can result in a significant degradation in the performance of our SPRINQL and the other baselines. Conversely, adding more sub-optimal data can enhance overall stability and lead to improved performance.

In the Figure 13 we show the learning curves with varied sizes for sub-optimal datasets. It can be seem that the overall performance tends to improve with more sup-optimal datasets. In particular, for the Ant task, our SPRINQL even fails to maintain stability when the sizes of sup-optimal datasets are low (10k-5k-1k). Moreover, while BC seems to show improvement with more sub-optimal data, the performance of DWBC and DemoDICE remains unchanged. This may be because these approaches rely on the assumption that expert demonstrations can be extracted from the sub-optimal data, which is not the case in our context.

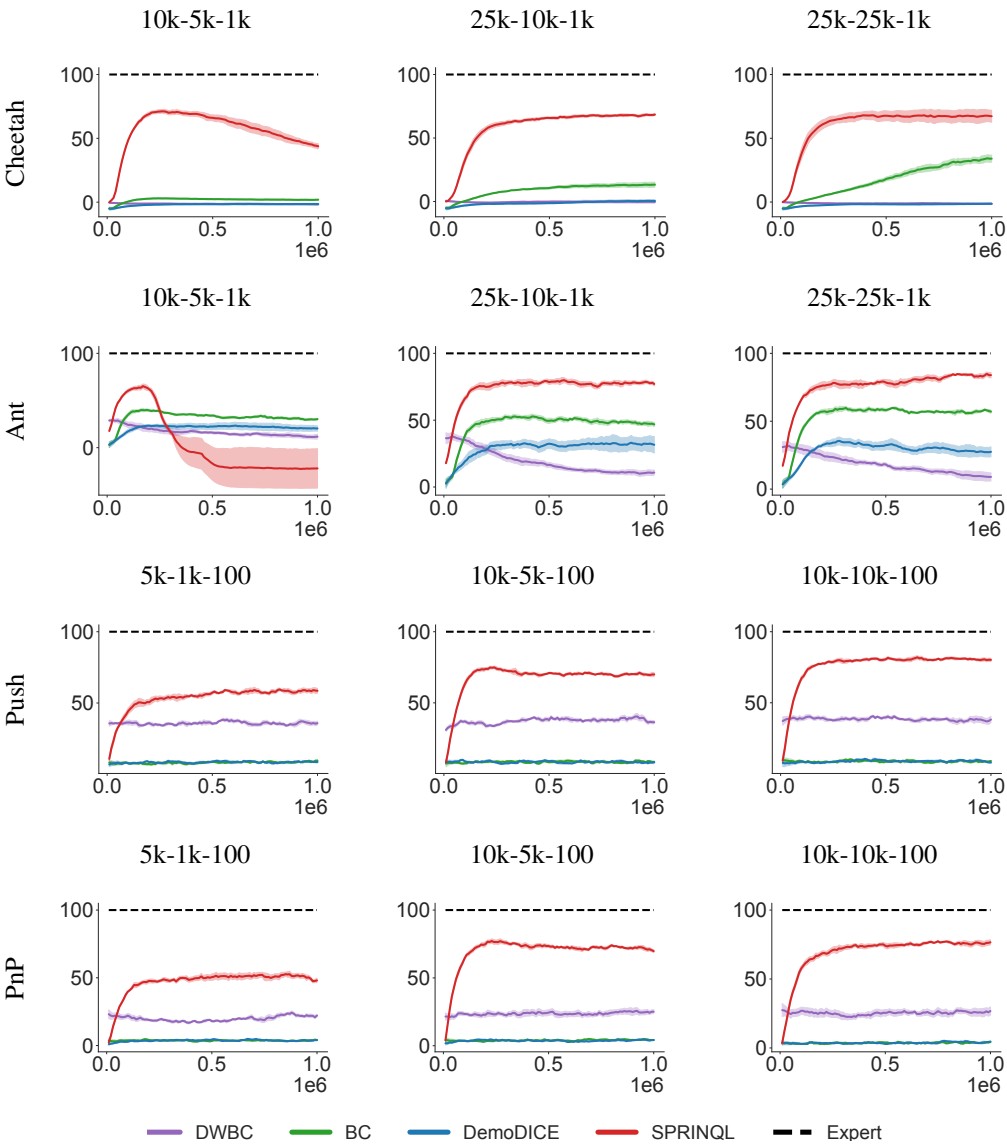

Figure 13: Evaluation curves with different sub-optimal-dataset size.

## C.6  Impact of the Conservative Term

In this experiment, we aim to answer (**Q5**) – *How does the conservative term help in our approach*? The Equation 8 introduce the conservative Q learning (CQL) term into our work. Here we also test three variants of our method from Section 4.3 and show the impact of CQL to the final performance. The experimental results are shown in Figure 14.

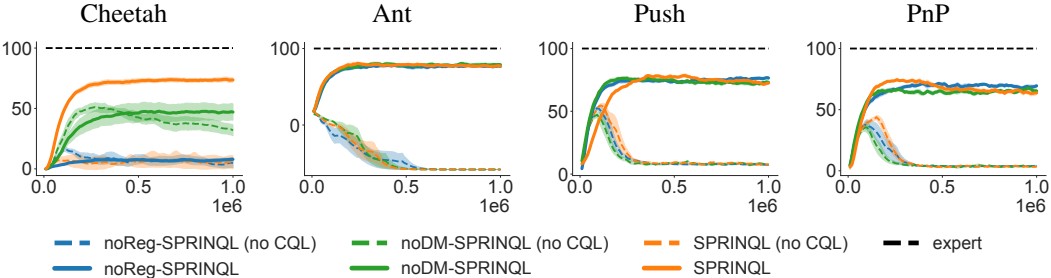

Figure 14: Performance of 3 variant with and without the CQL term in four different environments accross two domains.

## C.7  Performance with Varying Number of Expertise Levels

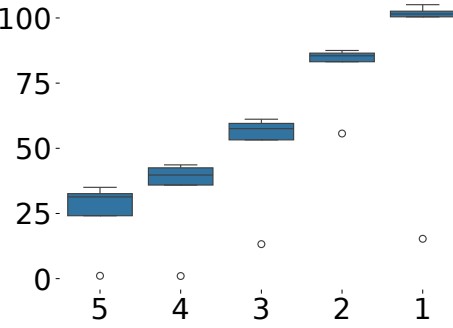

Figure 15: Average returns of the 5 *HalfCheetah* datasets (one expert and four sub-optimals).

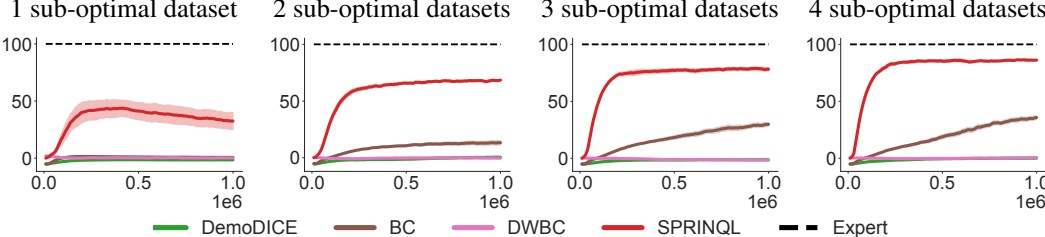

Figure 16: Experiment results for different numbers of sub-optimal datasets. The learning curves are calculated by mean with shaded by the standard error of 5 data seeds.

We provide an experiment to answer (**Q6**) - *How does increasing N (the number of expertise levels) affect the performance of SPRINQL?* We assess our algorithm with different numbers of expertise levels to investigate how adding or removing sub-optimal expertise levels influences performance. Specifically, we keep the same expert dataset comprising 1 expert trajectory and conduct tests with 1 to 4 sub-optimal datasets from the Cheetah task. Details of the average returns of the five datasets are reported in Fig. 15. In this context, SPRINQL outperforms other algorithms in utilizing non-expert

demonstrations. Furthermore, BC successfully learns and achieves performance comparable to the performance of SPRINQL with 2 dataset, while DemoDICE and DWBC struggle to learn. The detailed results are plotted in Figure 16. In Section (C.2) below, We particularly delve into the situation of having two datasets (one expert and one sub-optimal), which is a typical setting in prior work.

## C.8 Ablation Study for the Preference-based Weight Learning

We provide this experiment to answer (**Q7**) – *Does the preference-based weight learning approach provide good values for the weights $w_i$?* To this end, we compare the performance of SPRINQL based on the weights determined by the preference-based methods described in the main paper (denoted as **auto W**), and the following weighting scenarios:

- **Uniform W:** We chose the weights $w_i$ uniformly over $[0, 1]$ as $\mathbf{w} = \{0.55, 0.35, 0.15\}$ with a ratio of approximately $10 : 7 : 4$.

- **Reduced W:** Starting from *Uniform W*, we reduced the weights of the non-expert data and tested the weights $\mathbf{w} = \{0.65, 0.2, 0.15\}$ with a ratio of approximately $10 : 3 : 2$.

- **Increased W:** Starting from *Uniform W*, we increased the weights of the non-expert data and chose $\mathbf{w} = \{0.4, 0.32, 0.28\}$ with a ratio of approximately $10 : 8 : 7$.

These weight vectors $\mathbf{w}$ are normalized to follow $w_1 > w_2 > \ldots > w_N$ and $\sum_{i \in [N]} w_i = 1$.

The comparison results are shown in Figure 17.

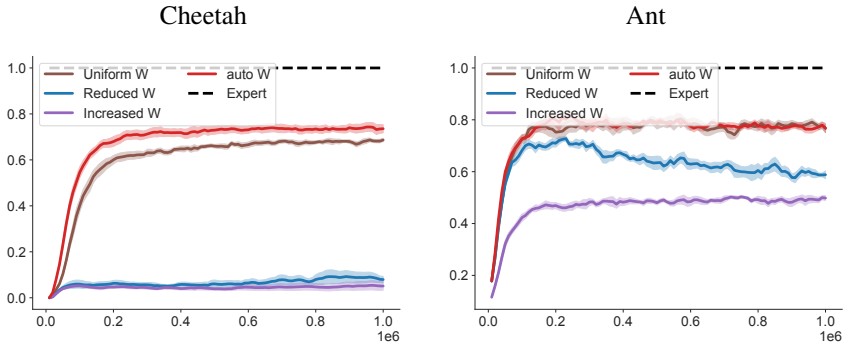

Figure 17: Experiment results for SPRINQL with different manual selection of weight $W$.

## C.9 D4RL mujoco dataset

In this section, we present additional experiments using the official D4RL dataset. Unfortunately, since our algorithm requires meaningful demonstrations, we exclude the Random dataset and are only able to test with $N = 2$ (Medium and Expert datasets). The detailed results are shown in Figure 18.

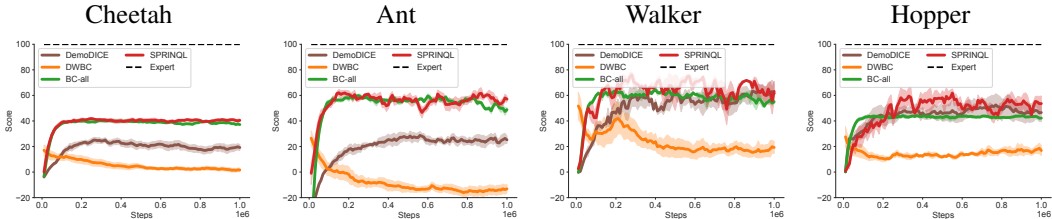

Figure 18: Performance in D4RL dataset with Medium (25,000 transitions) and Expert(1,000 transitions). The results are reported from 5 seeds per method.

## C.10 Reward Recovering

In this experiment, we want to answer (**Q8**) - *How does SPRINQL perform in recovering the ground-truth reward function?* Compared to BC-based algorithms, one notable advantage of Q-learning based algorithms is their ability to recover the reward function. Here, we present experiments demonstrating reward function recovery across five MuJoCo tasks, comparing recovered rewards to the actual reward function. To achieve this, we introduce increasing levels of random noise to the actions of a trained agent and observe its interactions with the environment. We collect the state, action, and next state for each trajectory, then predict the recovered reward and compare it to the true reward from the environment. For the sake of comparison, we include **noReg-SPRINQL**, which can be considered an an adaption of IQ-learn [13] to our setting, and **noDM-SPRINQL**, which is in fact an adaption of T-REX to our offline setting.

Comparison results are presented in Figure 19. We observe a linear relationship between the true and predicted rewards for SPRINQL across all testing tasks, whereas the other approaches fail to return correct relationships for some tasks.

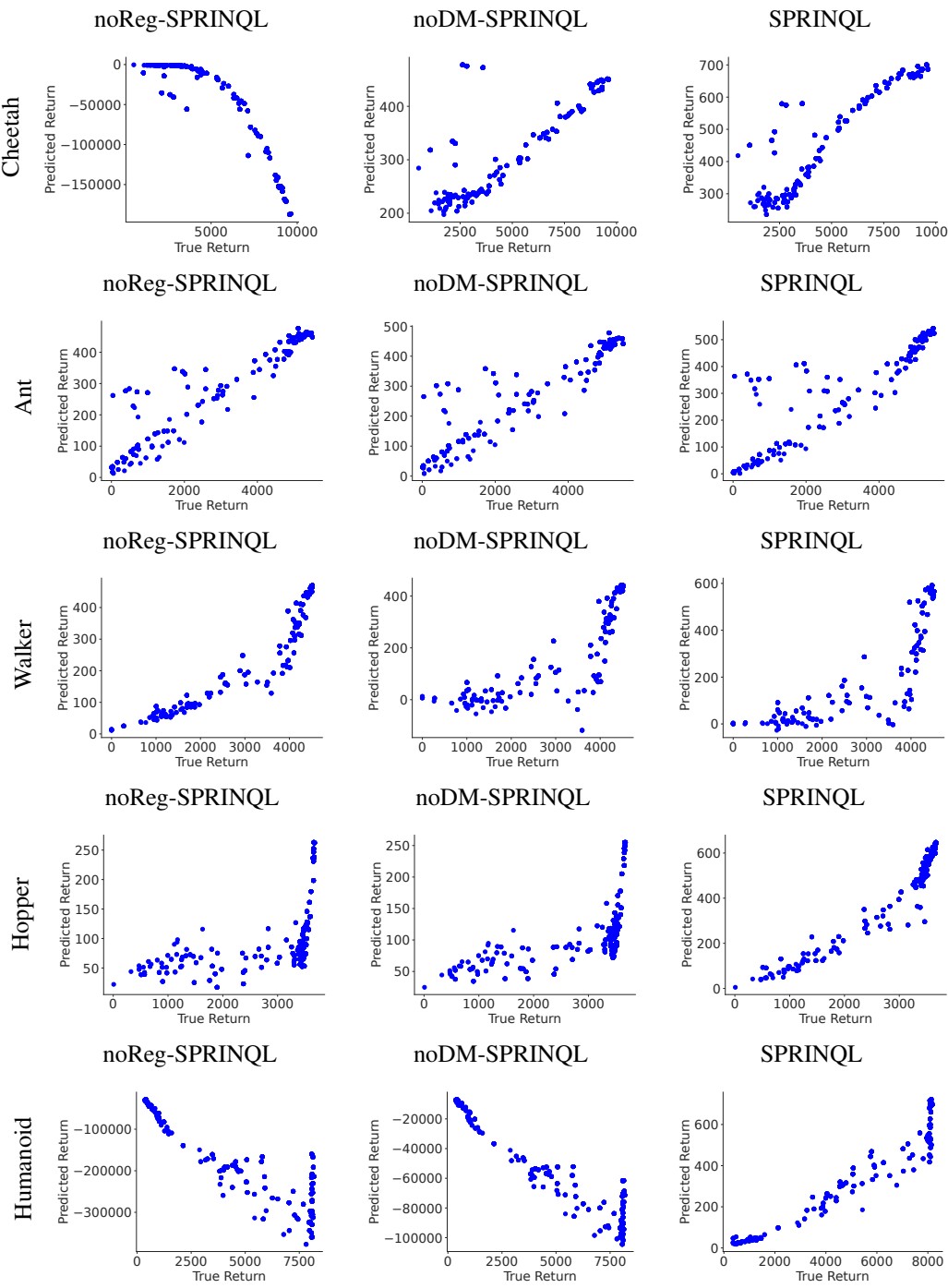

Figure 19: Recovered return and the true return of five Mujoco environments.

## C.11 Reference Reward Distribution

From the experiment reported in Table 1, we plot the distributions of the reward reference values in Figure 20, where the $x$-axis shows the level indexes and the $y$-axis shows reward values. The rewards seem to follow desired distributions, with larger rewards assigned to higher expertise levels. Moreover, rewards learned for expert demonstrations are consistently and significantly higher and exhibit smaller variances compared to those learned for sub-optimal transitions.

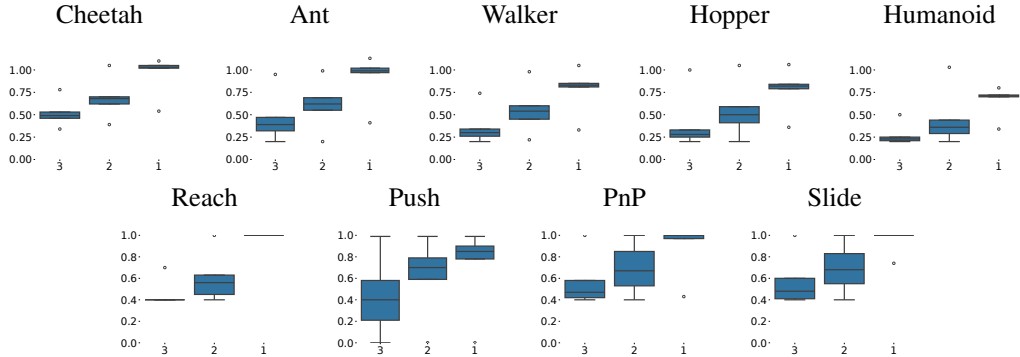

Figure 20: Whisker plots illustrate the reward reference distribution of three datasets of each environment for one seed.

## C.12 Different alpha ablation study

As our objective is a combination of two terms with a balancing parameter $\alpha$, we conduct additional experiments to evaluate the performance of our method across a range of $\alpha$ values. The detailed results are presented in Figure 21. Overall, the results indicate that $\alpha$ can be selected within the range of 0.1 to 10 to achieve good performance.

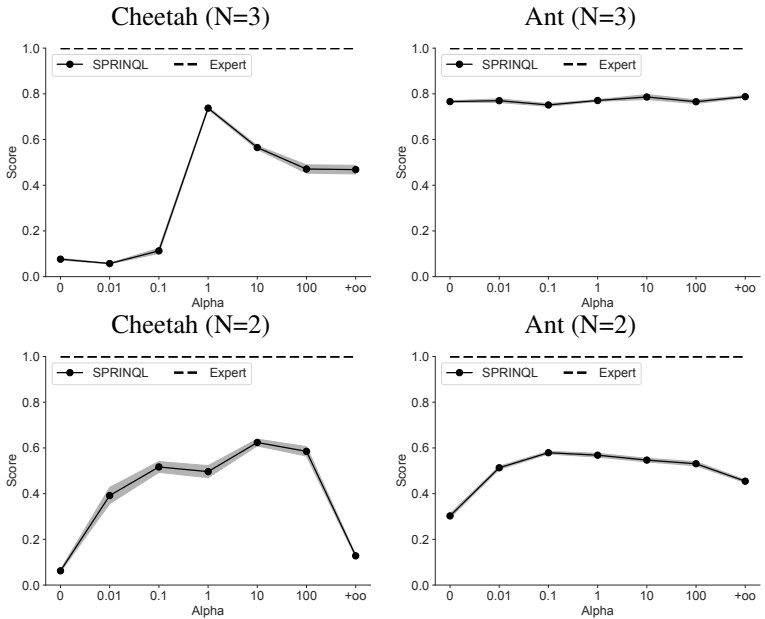

Figure 21: Performance of SPRINQL in different $\alpha$. The results are reported from 5 seeds per $\alpha$ value.

