# OpenReview forum: "SPRINQL: Sub-optimal Demonstrations driven Offline Imitation Learning"
_NeurIPS.cc/2024/Conference — NeurIPS 2024 poster_

### Official Review · Reviewer_qfkX · 2024-07-12

**Soundness:** 2
**Presentation:** 3
**Contribution:** 3
**Rating:** 5
**Confidence:** 4

**Summary:**

This paper considers the problem of offline imitation learning from sub-optimal demonstrations. The paper proposes the SPRINQL algorithm, which automatically weights the demonstrations, regularizes the loss with a reference reward function, and then performs inverse soft-Q learning. Experimental results show that the proposed method achieves better performance compared with baselines. Extensive ablations are also provided.

**Strengths:**

The paper is overall well written. The experimental results are strong, and the ablation studies are extensive. It is impressive to see the proposed method improves its performance with more sub-optimal demonstrations provided.

**Weaknesses:**

1. Some parts of the proposed approach are unclear to me. Please see questions 1-4.

2. The influence of the reward regularization term is not completely studied. Please see question 5.

**Questions:**

1. In Equation (3), a modified objective is presented. However, it seems that this objective depends largely on the quality of weights, and it does not necessarily recover the ground truth reward function nor the optimal policy. Given that the weight is chosen with a heuristic, what kind of guarantee can the authors give about the converged policy?

2. Following the first question, in the experiments, seems that the proposed algorithm always outperforms BC-E because of the lack of expert demonstrations. However, I am curious to see what will happen given sufficient expert demonstrations.

3. In equations above line 162, what if $(s, a) = (s', a')$, given that it is possible for two stochastic policies to choose the same action in the same state?

4. Line 173, regarding "prior research", could the authors add some citations?

5. It seems that the weight for reward regularization $\\alpha$ is an important hyperparameter. Will the performance of the proposed method change a lot with different $\\alpha$? In the extreme case, what if we learn a policy with RL using reward function $\\bar r$?

**Limitations:**

The authors discuss the limitations adequately in the last Section.

---

> ### Author Rebuttal · Authors · 2024-08-07
>
> We thank the reviewer for carefully reading our paper and for the constructive feedback.
>
> ---
>
> > In Equation (3), a modified objective is presented. However, it seems that this objective depends largely on the quality of weights, and it does not necessarily recover the ground truth reward function nor the optimal policy. Given that the weight is chosen with a heuristic, what kind of guarantee can the authors give about the converged policy?
>
> Thank you for the comment. In our setting, we have datasets with different levels of sub-optimality, and the amount of expert data is limited, making it generally impossible to recover the ground truth rewards or the optimal policy. What we can infer from equation (4) is that if $\alpha = 0$, then the algorithm will converge to $\rho^U$, which is a union of occupancy measures of different expert levels. If $\alpha$ is much larger than 1, the algorithm will return a policy optimal for the reference rewards (obtained by a preference-based approach). Our experiments show that neither extreme case works well (Section 4.3). In our experiments, we simply choose $\alpha = 1$, which we find reasonable to balance between the two extremes. In theory, the objective in equation (4) will generally return a policy that lies between the one given by matching the unioned occupancy measures and the one given by the reference rewards.
>
> To support our response above, we have conducted additional experiments (shown in Figure 3 of the attached 1-page PDF) testing our algorithm with varying $\alpha$ and computed the normalized score. These experiments generally show that $\alpha$ can be chosen between $0.1$ to $10$ to achieve good performance.
>
> We hope this addresses some of your concerns about the role of $\alpha$. If you have any other comments or questions regarding this, we would be happy to address them.
>
> > Following the first question, in the experiments, seems that the proposed algorithm always outperforms BC-E because of the lack of expert demonstrations. However, I am curious to see what will happen given sufficient expert demonstrations.
>
> It is well known that as we increase the number of expert demonstrations, the performance of BC-E will gradually improve and eventually surpass all other approaches (prior studies have already shown that BC performs very well with sufficient data). However, it is not feasible to have many expert trajectories and if we do have enough trajectories, there is no need for optimized offline imitation learning approaches. In Section C.4 and C.5 (appendix) of the experiments, we increased the amount of expert data and sub-optimal data to observe the effect. However, the focus was **not** on providing “sufficient” data to align with our initial problem setting and motivation.
>
> > In equations above line 162, what if $(s,a) = (s',a')$, given that it is possible for two stochastic policies to choose the same action in the same state?
>
> Thank you for the observation. In our datasets,  if a (state, action) pair appears in an expert dataset, it will be excluded from any lower-level expert dataset. We will clarify this point.
>
> > Line 173, regarding "prior research", could the authors add some citations?
>
> This has been investigated in the prior IQ-learn paper and some subsequent works. We will add relevant citations for this. Thank you for the comments.
>
> > It seems that the weight for reward regularization $\alpha$ is an important hyperparameter. Will the performance of the proposed method change a lot with different $\alpha$? In the extreme case, what if we learn a policy with RL using a reward function
> $\bar{r}$?
>
> The two extreme cases $(\alpha = 0$ and $\alpha \gg 1$) have already been considered and tested; they correspond to noDM-SPRINQL and NoReg-SPRINQL in the ablation study in Section 4.3. These two extreme cases are generally outperformed by our main algorithm. In Figure 3 of the attached 1-page PDF, we provided additional experiments showing the performance of our algorithm with varying $\alpha$.

---

> > ### Comment · Reviewer_qfkX · 2024-08-09
> >
> > I want to thank the authors for their reply. The reply addresses my concerns, and the additional experiments provided are informative. However, one of my remaining concerns is that even with sufficient expert demonstrations, seems the proposed algorithm still cannot recover the expert policy. The authors claim that the proposed algorithm works better when "the amount of expert data is limited", but it is difficult to quantify how much data is "limited" and how much is "sufficient". Overall, I am positive about the paper, and I am keeping my original score.

---

> ### Author Response · Authors · 2024-08-13
>
> Thank you to the reviewer for carefully reading our responses, offering further feedback, and maintaining a positive view of our paper.
>
> We would like to clarify that, in our context, "sufficient" expert data refers to the quantity required by offline RL methods (e.g., BC, IQ Learn) that rely exclusively on expert data to achieve near-expert performance. Conversely, "insufficient" data would mean that these offline RL methods are unable to effectively learn using only expert demonstrations. These thresholds can vary depending on the environment.
>
> In our main experiments, we intentionally kept the amount of expert demonstrations "insufficient" (otherwise, standard imitation methods like IQ-Learn or BC could perform well with just expert data) to demonstrate how our method can leverage suboptimal data to support learning when expert data alone is insufficient.
>
> To further support our response, we conducted a few additional experiments (see table below) where the number of expert demonstrations was increased from "insufficient" to "sufficient". The results generally show that our method performs well with a significantly smaller number of expert demonstrations, remains competitive when the number of demonstrations is sufficient, and can achieve expert-level performance in the Ant environment.
>
>
> |||Cheetah||||Ant|||
> |:-----------:|:-----------:|:-----------:|:-----------:|:-----------:|:-----------:|:-----------:|:-----------:|:-----------:|
> | Expert Transitions    | 1k   | 5k   | 10k   | 25k   | 1k   | 5k   | 10k   | 25k   |
> | DemoDICE    |   0.4 $\pm$ 2.0  | 12.2 $\pm$ 5.2  | 40.0 $\pm$ 10.5  | 71.5 $\pm$ 10.8  | 31.7 $\pm$ 8.9  | 79.4 $\pm$ 6.6  | 81.7 $\pm$ 5.2  | 84.0 $\pm$ 4.9 |
> | SPRINQL    | 73.6 $\pm$ 4.3  | 86.7 $\pm$ 3.5  | 85.9 $\pm$ 4.1  | 88.6 $\pm$ 3.4  | 77.0 $\pm$ 5.6  | 88.6 $\pm$ 4.3  | 96.6 $\pm$ 8.3  | 96.7 $\pm$ 9.5  |
>
> *We hope the responses above help address the remaining concerns you may have!*

---

> > ### Comment · Reviewer_qfkX · 2024-08-14
> >
> > Thank the authors for their reply! However, these experimental results do not reply directly to my question. My question was whether, with sufficient expert demonstrations, could the proposed algorithm recover the expert policy or not. Could the authors please clarify this, or provide the comparison with BC-E with a different number of expert transitions?

---

> > > ### Author Response · Authors · 2024-08-14
> > >
> > > Thank you for your feedback. We would like to clarify that when there is sufficient expert data available:
> > >
> > > - Achieving expert-level performance consistently with our approach may not always be possible, as part of our objective involves learning from sub-optimal data.
> > > - Methods such as DWBC, Demo-DICE, or our proposed approach are unnecessary. Pure offline imitation learning techniques, like BC or IQ-Learn, should be preferred.
> > >
> > > Below, we provide a quick set of results on the Cheetah and Ant environments, including BC-E for comparison.
> > >
> > > | |  | Cheetah |  | | | Ant | | |
> > > |:----------:|:----------:|:----------:|:----------:|:----------:|:----------:|:----------:|:----------:|:----------:|
> > > |   Expert Transitions   |   1k   |   5k   |   10k   |   25k   |   1k   |   5k   |   10k   |   25k   |
> > > |   BC-E   |   -3.2 $\pm$ 0.9    |   1.9 $\pm$ 3.0    |   44.5 $\pm$ 10.6    |   79.8 $\pm$ 6.2    |   6.4 $\pm$ 19.1    |   9.0 $\pm$ 6.6  |  50.5 $\pm$ 7.2   |   86.7 $\pm$ 3.9    |
> > > | DemoDICE    |   0.4 $\pm$ 2.0  | 12.2 $\pm$ 5.2  | 40.0 $\pm$ 10.5  | 71.5 $\pm$ 10.8  | 31.7 $\pm$ 8.9  | 79.4 $\pm$ 6.6  | 81.7 $\pm$ 5.2  | 84.0 $\pm$ 4.9 |
> > > | SPRINQL    | 73.6 $\pm$ 4.3  | 86.7 $\pm$ 3.5  | 85.9 $\pm$ 4.1  | 88.6 $\pm$ 3.4  | 77.0 $\pm$ 5.6  | 88.6 $\pm$ 4.3  | 96.6 $\pm$ 8.3  | 96.7 $\pm$ 9.5  |
> > >
> > >
> > > *We hope the above response resolves your remaining concerns.*

---

### Official Review · Reviewer_vP7z · 2024-07-12

**Soundness:** 4
**Presentation:** 4
**Contribution:** 4
**Rating:** 7
**Confidence:** 4

**Summary:**

This paper considers the problem of imitation learning on suboptimal demonstrations given coarse trajectory expertise rankings. It introduces SPRINQL, a method combining inverse soft-Q learning, conservative Q learning, and reward learning from ranked trajectory preferences. SPRINQL estimates a reference reward function given trajectory expertise rankings, and extends inverse soft-Q learning to the suboptimal demonstration setting by adding reward regularization and weights based on the estimated reference reward. The paper shows that the SPRINQL objective is tractable with a unique solution over $Q, \pi$. Experiments show that SPRINQL outperforms prior methods on suboptimal trajectories in simulated environments. Ablations justify each component in SPRINQL and demonstrate robustness across a range of dataset sizes and number of suboptimal datasets.

**Strengths:**

- SPRINQL works with a number of suboptimal demonstrations, with reasonable assumptions such as access to trajectory expertise rankings.
- The main objective is shown to admit a unique saddle point solution over $Q, \pi$ and is tractable for optimization.
- Experiments show that SPRINQL outperforms relevant offline imitation learning baselines on suboptimal trajectories in multiple simulated locomotion and manipulation environments.
- Extensive ablations demonstrate that the method works with a range of numbers/sizes of suboptimal datasets, and justify each component in SPRINQL.

**Weaknesses:**

- Suboptimal trajectories are sourced from adding random noise to expert actions. While this is a reasonable proxy, it would be more convincing to see them generated from under-trained RL policies instead.
- It would be great to see SPRINQL work on image-based environments.

**Questions:**

- How does SPRINQL perform on suboptimal trajectories generated from under-trained RL policies, instead of rollouts from noisy expert actions?

**Limitations:**

- SPRINQL assumes access to trajectory expertise rankings.

---

> ### Author Rebuttal · Authors · 2024-08-07
>
> We thank the reviewer for reading our paper and the positive feedback.
>
> ---
>
> > Suboptimal trajectories are sourced from adding random noise to expert actions. While this is a reasonable proxy, it would be more convincing to see them generated from under-trained RL policies instead. How does SPRINQL perform on suboptimal trajectories generated from under-trained RL policies, instead of rollouts from noisy expert actions?
>
> We thank the reviewer for the comment. While we believe adding noise is a practical setting, we agree that evaluating undertrained policies is also reasonable. In response, we have conducted additional experiments in this setting using the standard D4RL dataset (see Figure 1 in the attached 1-page PDF). These results generally show that our approach remains competitive or superior compared to other baselines.
>
> > It would be great to see SPRINQL work on image-based environments.
>
> The reviewer's point is well noted. Unfortunately, to the best of our knowledge, there are no image-based datasets available for offline imitation learning with sub-optimal data. We have utilized the accepted benchmarks from the literature in offline imitation learning for our evaluation and benchmarking. We hope to consider this for future work.

---

> > ### Comment · Reviewer_vP7z · 2024-08-11
> >
> > Thank you for the additional experiments. Overall I think this is a good paper and I would like to keep my score of Accept.

---

> > > ### Author Response · Authors · 2024-08-12
> > >
> > > Thank you to the reviewer for reading our responses and maintaining a positive view on our paper!

---

### Official Review · Reviewer_78nP · 2024-07-14

**Soundness:** 3
**Presentation:** 3
**Contribution:** 3
**Rating:** 7
**Confidence:** 3

**Summary:**

This paper presents SPRINQL (Sub-oPtimal demonstrations driven Reward regularized INverse soft Q Learning), a novel algorithm for offline imitation learning with multiple levels of expert demonstrations. SPRINQL leverages distribution matching and reward regularization to effectively leverage the mixture of non-expert data and expert data to enhance learning. An extensive evaluation is conducted to assess SPRINQL, answering eight well-defined questions regarding the capability and performance of SPRINQL.

**Strengths:**

+ The proposed technique presents an improvement for learning from mixed-expert demonstrations and is backed by theory.
+ The paper is very well-written and contains sufficient detail to understand the key proposed novelties.
+ The evaluation (including those in the appendix) is extensive and presents strong support for the proposed algorithm.

**Weaknesses:**

- The authors mention that this setting (having several sets of data labeled with relative quality scores) is more general than two quality levels: expert and sub-optimal. Could the authors provide further support for this claim? I would argue that having additional quality annotations makes this setting more niche.
- Table 1 contains a very large evaluation but there isn't much commentary on the different trends between baselines and SPRINQL.
Comment: The example in the introduction does not seem complete.

**Questions:**

Please address the weaknesses above.

**Limitations:**

Yes

---

> ### Author Rebuttal · Authors · 2024-08-07
>
> We thank the reviewer for the positive feedback and the insightful comments.
>
> ---
>
> > The authors mention that this setting (having several sets of data labeled with relative quality scores) is more general than two quality levels: expert and sub-optimal. Could the authors provide further support for this claim? I would argue that having additional quality annotations makes this setting more niche.
>
> First, we would like to restate that in our setting, there are datasets labeled with different levels of expertise, but relative scores are not available. If there are only two expertise levels, our setting corresponds to the prior work's two-quality level setting. Additionally, our setting, which includes several datasets of different expertise levels, stems from practical observations, such as from projects on treatment optimization we are running, where we have demonstrations from doctors with varying levels of experience. One can also think of applications in autonomous driving, where demonstrations come from drivers of different experience levels, or demonstrations from large language models of different qualities.
>
> > Table 1 contains a very large evaluation but there isn't much commentary on the different trends between baselines and SPRINQL. Comment: The example in the introduction does not seem complete.
>
> We did not include detailed commentary in our paper due to space constraints. The reviewer’s points are well noted. We will add more discussion to analyze the results reported in Table 1. Additionally, we will elaborate on the robotic manipulation example in the introduction and provide another example on treatment optimization to further support our motivation.

---

> > ### Comment · Reviewer_78nP · 2024-08-11
> >
> > Dear Authors,
> >
> > Thank you for your response. After reading all the reviews and their respective replies, I have decided to maintain my score of Accept.

---

> > > ### Author Response · Authors · 2024-08-12
> > >
> > > Thank you to the reviewer for reading our responses and maintaining a positive view of our work!

---

### Official Review · Reviewer_g6FW · 2024-07-27

**Soundness:** 2
**Presentation:** 3
**Contribution:** 2
**Rating:** 5
**Confidence:** 3

**Summary:**

This work presents an offline imitation learning (IL) method that leverages both expert and suboptimal demonstrations. Distinct from previous methods, this approach assumes access to suboptimal demonstrations across a spectrum of performance levels, with a known ranking to the learners.
To utilizes suboptimal demonstations with various levels, this work suggest SPRINQL (Sub-oPtimal demonstrations driven Reward regularized INverse Q-Learning), a reward reguarized inverse soft Q-learning method for the offline IL problem.
The evaluation of this method is conducted using the Mujoco and Panda-gym environments, employing a small dataset of expert demonstrations and a larger collection of demonstrations generated by introducing varying levels of noise to expert actions.

**Strengths:**

This work aims to extend the domain of offline imitation learning by addressing the challenge of leveraging imperfect demonstrations with multiple suboptimality levels, presenting a novel and intriguing approach. Drawing on foundational principles from prior research, notably IQ-Learn, this study demonstrates theoretically that the proposed extension, SPRINQL, retains the advantageous properties of IQ-Learn.
Furthermore, SPRINQL adeptly incorporates sophisticated concepts such as the Bradley-Terry model and Conservative Q-Learning (CQL) into its training framework. The experimental outcomes are notable, with SPRINQL achieving surprisingly effective results that surpass previous baselines, which were unable to successfully imitate expert performance using the datasets used in this work.

**Weaknesses:**

Significant concerns with this submission are the questionable reporting on the performance of baseline methods and issues related to reproducibility.
The experimental results presented are not entirely convincing, as the performance of the baselines substantially deviates from those reported in their original papers.
Specifically, even in the case with N=2 suboptimality levels, suitable baselines such as TRAIL, DemoDICE, and DWBC fail to outperform even basic BC-all in nearly all the Mujoco tasks listed in Table 3 (Appendix).
The authors need to clarify the reasons behind this anomaly.
To enhance the trustfulness of experiments, it would be beneficial for the authors to release the datasets utilized in their experiments or to use standardized datasets such as D4RL or RL-unplugged. This would allow for a more reliable validation of the results and facilitate comparisons with other methods.

**Questions:**

- Could the authors clarify the motivation of employing reference rewards?
- Moreover, could you elucidate why reference rewards are utilized as a regularization mechanism rather than directly applying RL using reference rewards?
- The problem setting appears closely aligned with preference-based learning, where optimality (or preference) levels are explicitly known. Could you delineate the distinctions between these two problem settings?
- Can you provide detailed justifications for why comparing the proposed method with IPL is deemed unsuitable?
It appears feasible to compare by segmenting each trajectory by suboptimality levels, then pairing segments from differing levels to facilitate IPL. Could this approach not be applied effectively?

[minor comments] reference [15] in line 398 from "NeurIPS 2024" to "NeurIPS 2023."

**Limitations:**

The authors addressed their limitations in Conclusion section in the main manuscript. They described that this work does not have any direct societal impacts in the Checklist #10, Broader Impacts.

---

> ### Author Rebuttal · Authors · 2024-08-07
>
> We thank the reviewer for carefully reading our paper and providing us with constructive feedback.
>
> ---
>
> > Significant concerns with this submission are the questionable reporting on the performance of baseline methods and issues related to reproducibility. The experimental results presented are not entirely convincing, as the performance of the baselines substantially deviates from those reported in their original papers...
>
> Our dataset, as well as all the code to reproduce the experiments, is already uploaded to an anonymized Google Drive folder and is ready to be made publicly available.
>
> **Poor performance of DemoDICE and DWBC:**
>
> In DemoDICE and DWBC papers, their dataset for N=2 contains:
> - Set 1: a small set of expert trajectories; and
> - Set 2: an unlabelled set of trajectories (which also contains expert trajectories).
>
> In our case, for N=2, our dataset contains:
> - Set 1: a small set of expert trajectories
> - Set 2: a large set of sub-optimal trajectories (no expert trajectories are present).
>
> The key difference is the presence of expert trajectories in Set 2. DemoDICE and DWBC learn only from expert trajectories in Set 1 and Set 2, and ignore the sub-optimal trajectories. Their performance is reliant on expert trajectories being present in Set 2. Since, in our case, we do not have any expert trajectories in the set of sub-optimal trajectories, their performance suffers as they must learn only from the set of labeled expert trajectories.
>
> Our approach on the other hand learns from the sub-optimal trajectories in Set 2 as well and hence can provide better performance.
>
> In summary, the two reasons for the poor performance of DemoDICE and DWBC are:
> - Our dataset does not contain expert trajectories in Set 2.
> - DemoDICE and DWBC learn from expert trajectories in Set 1 and Set 2 and ignore sub-optimal trajectories.
>
> To support the above explanation, we have conducted additional experiments using a standard dataset (i.e., D4RL - see Figure 1 in the attached 1-page PDF), where sub-optimal data does not contain expert demonstrations. The results show that our algorithm and BC on average perform better than DemoDICE and DWBC.
>
> **Poor performance of TRAIL:**
>
> For TRAIL, they do not assume to have expert demonstrations in the sub-optimal data but require a very large set of sub-optimal data to learn the environmental dynamics. For instance, in the TRAIL paper, they use 1 million sub-optimal transitions, compared to (25000+10000) in our approach.
>
> **Comparison to BC:**
>
> TRAIL, DemoDICE, and DWBC cannot outperform BC if their strong assumptions -- expert trajectories in Set 2 for DemoDICE and DWBC, a large number of sub-optimal trajectories in Set 2 --  do not hold. Please refer to our additional experiments reported in Figure 1 of the 1-page PDF.
>
> ---
>
> > Could the authors clarify the motivation of employing reference rewards?
>
> Distribution matching (DM) term commonly used in offline imitation learning leads to overfitting when there is only a small amount of expert data. Therefore, we utilized the reference rewards in a regularization form to ensure that expert demonstrations are assigned higher rewards than non-expert ones, enhancing the effectiveness of the DM. Our experiments in Section 4.3 clearly demonstrate this aspect.
>
> > Moreover, could you elucidate why reference rewards are utilized as a regularization mechanism rather than directly applying RL using reference rewards?
>
> We utilize it as a regularization mechanism, as Distribution Matching (DM) is essential for good performance in imitation learning settings.  This is observed in our ablation study of Section 4.3, where the noDM-SPRINQL (directly utilizing RL with the reference rewards) performs worse than our main algorithm, especially for challenging tasks such as  Humanoid.
>
> > The problem setting appears closely aligned with preference-based learning, where optimality (or preference) levels are explicitly known. Could you delineate the distinctions between these two problem settings?
>
> The distinction here is that, in preference-based learning, trajectories are ranked by experts, so given a pair of trajectories, we know which one is more preferred. In our work, information comes from non-experts as well. The data come from experts of different levels, so we can only ensure that the expected rewards from higher-level experts are better. Unlike in preference-based learning, in our case, a specific trajectory from a higher-level expert is not necessarily better or preferred than one from a lower-level expert. Therefore, our setting is different.
>
> > Can you provide detailed justifications for why comparing the proposed method with IPL is deemed unsuitable? It appears feasible to compare by segmenting each trajectory by suboptimality levels, then pairing segments from differing levels to facilitate IPL. Could this approach not be applied effectively?
>
> As discussed above, IPL is a preference-based approach, and due to differences in assumptions about how the data is obtained, we did not find the comparison suitable. However, reviewers' point is well noted and we address this in two ways:
> - IPL is quite similar to our noDM-SPRINQL (which learns Q functions from ranked data using the Bradley-Terry model). The key difference is that IPL learns the reward and Q function simultaneously, while noDM-SPRINQL learns them sequentially. Section 4.3 shows that noDM-SPRINQL is not effective in our setting.
> - We have now provided additional experiments (Figure 2 in the 1-page PDF) comparing our method with IPL, which generally show that our algorithm performs significantly better.

---

> > ### Comment · Reviewer_g6FW · 2024-08-11
> >
> > Thank you for addressing my concerns and providing thoughtful comments.
> >
> > After reviewing your responses and the additional experimental results on D4RL, I find the results more convincing, even though the improvement over BC-all is slightly marginal in this context. Despite this, I believe the initial experimental setup proposed by the authors more effectively demonstrates the advantages of the suggested method.
> >
> > Hence, I have raised my assessment and am now leaning slightly towards accepting this paper.

---

> > > ### Author Response · Authors · 2024-08-12
> > >
> > > We are grateful to the reviewer for reading our responses and for the positive evaluation of our work. We will update our paper, incorporating your constructive feedback and suggestions

---

### Author Rebuttal · Authors · 2024-08-07

We thank the reviewers for carefully reading our paper and providing constructive feedback. We have made efforts to address all the points raised. Please find here, some key major points we wish to emphasize:

**Comparison to DWBC, DemoDICE, TRAIL:** Our key contribution is an offline imitation learning approach that learns not only from:
- Expert trajectories; but also, from
- sub-optimal trajectories.

On the other hand,
- DWBC and DemoDICE focus only on learning from expert trajectories.
- DWBC and DemoDICE have an unlabelled set of trajectories, but they identify expert trajectories from the unlabelled set and ignore the sub-optimal trajectories.
- TRAIL learns from sub-optimal trajectories but requires a significant number (1 million) of transitions compared to our approach (35000).

We have provided additional experiments using D4RL (Figure 1 in the 1-page pdf), that demonstrate the superiority of our approach. We also provide experiments to compare against IPL (Preference-based learning) in Figure 2 of the 1-page pdf.

**Motivation for Multiple experience levels:** Additionally, our setting, which includes several datasets of different expertise levels, stems from practical observations, such as from projects on treatment optimization we are running, where we have demonstrations from doctors with varying levels of experience. One can also think of applications in autonomous driving, where demonstrations come from drivers of different experience levels, or demonstrations from large language models of different qualities.


**Utilizing under-trained policies instead of noise to create sub-optimal datasets:**  We have conducted these additional experiments in this setting using the standard D4RL dataset (see Figure 1 in the 1-page PDF). These results generally show that our approach remains competitive or superior compared to other baselines.

**Impact of the reward regularization weights on the performance of the proposed algorithm:** We have conducted additional experiments (shown in Figure 3 of the attached 1-page PDF) testing our algorithm with varying $\alpha$.

___

We have provided specific responses to individual reviewer questions below and happy to answer any other questions reviewers/AC may have.

---

### Decision · Program_Chairs · 2024-09-25

**Decision:**

Accept (poster)

**Comment:**

This paper proposes an offline imitation learning (IL) method that leverages both expert and suboptimal demonstrations. This method requires access to suboptimal demonstrations across a spectrum of performance levels, with a known ranking to the learners. Experiments  reported in the paper show that the proposed solution outperforms prior methods on suboptimal trajectories in simulated environments. Ablations justify each component in the proposed solution.